# Aligning Evaluation with Clinical Priorities: Calibration, Label Shift, and Error Costs

**Gerardo A. Flores**[*]
MIT

**Alyssa H. Smith**
Northeastern University

**Julia A. Fukuyama**
Indiana University

**Ashia C. Wilson**
MIT

## Abstract

Machine learning-based decision support systems are increasingly deployed in clinical settings, where probabilistic scoring functions are used to inform and prioritize patient management decisions. However, widely used scoring rules, such as accuracy and AUC-ROC, fail to adequately reflect key clinical priorities, including calibration, robustness to distributional shifts, and sensitivity to asymmetric error costs. In this work, we propose a principled yet practical evaluation framework for selecting calibrated thresholded classifiers that explicitly accounts for uncertainty in class prevalences and domain-specific cost asymmetries. Building on the theory of proper scoring rules, particularly the Schervish representation, we derive an adjusted variant of cross-entropy (log score) that averages cost-weighted performance over clinically relevant ranges of class balance. The resulting evaluation is simple to apply, sensitive to clinical deployment conditions, and designed to prioritize models that are both calibrated and robust to real-world variations.

## 1 Introduction

The field of medicine increasingly relies on machine learning tools for clinical decision support. For both diagnostic and prognostic applications, probabilistic scoring functions inform and prioritize decisions about patient management. As such, it is critical that these scoring functions reflect clinical priorities. We propose three principles that scoring functions used for clinical purposes should satisfy as closely as possible. First, scoring functions should be adapted to account for the *known label shifts* that commonly arise between development and deployment environments. Because medical datasets are often deliberately rebalanced during development to improve sensitivity to rare outcomes, standard metrics computed under these distributions can overestimate performance in deployment, where class prevalences are highly skewed. Second, the scores returned by scoring functions should be sensitive to the relative *cost of errors* that are clinically significant, such as the trade-off between the cost of misdiagnosis and the cost of failing to diagnose in any given setting. This supports patient-centered care by enabling the classifier's sensitivity to be calibrated to human feedback, rather than presuming a fixed normative standard. Third, scores should be *calibrated* to ensure that their outputs align with a common interpretive frame, enabling practitioners to form consistent and reliable expectations about risk and outcomes.

This work focuses on *evaluation*; specifically, we examine how the field of medical machine learning assesses and compares scoring functions and the extent to which current evaluation practices reflect these aforementioned clinical priorities. We begin by showing that neither of the most commonly used metrics, accuracy and AUC-ROC, adequately captures all three priorities outlined above. Each abstracts away some considerations that are critical for clinical decision-making.

We structure the paper as follows. We first examine *accuracy* and its variants, as these remain the most widely used evaluation metrics for classification tasks. Accuracy evaluates each decision

---

[*]Corresponding author: `nullset@mit.edu`.
   Code available at `https://github.com/nullset-mit/epamnb`

39th Conference on Neural Information Processing Systems (NeurIPS 2025).

independently and measures the overall proportion of correct predictions, abstracting away critical application-specific considerations such as *class imbalance* and *asymmetric error costs*. While this abstraction offers a form of neutrality, it obscures important aspects of clinical deployment, where decision thresholds must often be adapted to reflect evolving prevalence rates or varying tolerances for false positives and false negatives. As several works have noted Bradley [1997], Provost and Fawcett [1997], Drummond and Holte [2006], accuracy fixes a single operating point and therefore fails to accommodate the flexibility required for robust clinical evaluation.

We then turn our attention to the Area Under the Receiver Operating Characteristic Curve (AUC-ROC), which is commonly viewed as a solution to the rigidity of accuracy, as it evaluates classifier performance across all possible thresholds. However, AUC-ROC measures the expected performance of the *ideally calibrated* version of a scoring function, not the actual, potentially *miscalibrated* outputs of a model. Moreover, it ties evaluation to a distribution over positive prediction rates that may not correspond to clinical contexts. These assumptions often lead AUC-ROC to overstate the real-world reliability of scoring functions, especially when calibration is imperfect or deployment conditions differ from development data.

Recent work in the fairness literature has explored calibration more directly [Kleinberg et al., 2017, Corbett-Davies et al., 2017, Hebert-Johnson et al., 2018], but the literature lacks broad consensus on best practices for how calibration interacts with varying cost structures. In particular, the definition of perfect calibration is widely agreed upon, but the correct way to measure degrees of miscalibration, taking into account label prevalence and asymmetric error costs, is not. As a consequence, the use of calibration-based metrics has lagged behind that of accuracy and AUC-ROC in clinical ML settings. To address these concerns regarding current evaluation practices, we propose adapting a framework from the weather forecasting and belief elicitation literature known as the Schervish representation Schervish [1989]. This framework shows that any proper scoring rule (a measure of calibration that does not require binning) can be represented as an integral over discrete cost-weighted losses, directly linking calibration to decision-theoretic performance. We extend this framework to the setting of label shift and asymmetric costs, and we average cost-sensitive metrics over a bounded range of class balances.

In summary, this work makes two main contributions. First, we introduce a framing of scoring rule design that centers clinical priorities, namely calibration, robustness to distributional shift, and sensitivity to error costs. Second, we propose an adaptable scoring framework based on adjusted log scores that reflects clinical needs. It accommodates uncertainty in class balance, asymmetric cost structures, and the requirement for calibrated predictions, thereby offering a more principled foundation for evaluating machine learning models in clinical decision support.

## 1.1 Problem Formulation

Given an input space $\mathcal{X}$ and binary label space $\{0, 1\}$, the standard goal of binary classification is to learn a decision rule that maps each input $x \in \mathcal{X}$ to a predicted label. To do so, a scoring function $s : \mathcal{X} \to \mathbb{R}$ is designed to assign a real-valued score to each input, which is then converted into a binary prediction by thresholding. For a threshold parameter $\tau \in \mathbb{R}$, the predicted label is defined as

$$\kappa(s(x), \tau) = \mathbf{1}_{(s(x) \geq \tau)},$$

where $\mathbf{1}_{(\cdot)}$ denotes the indicator function, equal to $1$ if the argument is true and $0$ otherwise. We denote the dataset by $\mathcal{D}_{\pi_0}$, consisting of input-label pairs $(x, y)$ drawn from an unknown distribution. We define the *empirical class prevalence* as $\pi_0 = \mathbb{P}_{\mathcal{D}_{\pi_0}}(y = 1)$, which represents the proportion of positive examples in the dataset and the possibly unknown *target* or deployment *class prevalence* as $\pi = \mathbb{P}_{\mathcal{D}_\pi}(y = 1)$. To formalize evaluation objectives, we introduce three additional elements: (1) A value function where $V(y, \kappa(s(x), \tau))$ specifies the utility or loss associated with predicting $\kappa(s(x), \tau)$ when the true label is $y$; and (2) a parameter $c \in (0, 1)$, which encodes the relative cost of false positives and false negatives and determines the threshold; and (3) a distribution $H$ over possible data-generating distributions $\mathcal{D}_\pi$, modeling uncertainty over the environment and potential distribution shifts. We denote odds multiplication by $a \otimes b \triangleq \frac{ab}{ab+(1-a)(1-b)}$. We denote the clipping operator by $\text{clip}_{[a,b]}(x) \triangleq \max(a, \min(b, x))$.

## 1.2 Related Work

We provide an extensive discussion of related literature which includes calibration techniques, robustness to distribution shift, cost-sensitive learning, visualization methods, and more in Appendix A. Here, we focus on the most directly relevant conceptual threads that motivate our approach and situate our contributions. A growing body of work in AI emphasizes that evaluation metrics must reflect deployment-specific decision contexts rather than relying on generic benchmarks that can obscure model behavior [Cruz Rivera et al., 2020, Liu et al., 2020]. This view aligns with long-standing traditions in decision theory, including Decision Curve Analysis (DCA) [Vickers and Elkin, 2006], which critiques metrics like AUC and the Brier score for lacking clinical relevance [Vickers and Holland, 2021, Assel et al., 2017].

The theory of proper scoring rules has similarly evolved from early formulations [Brier, 1950, Good, 1952] to rich characterizations linking scores to decision-theoretic utility [Schervish, 1989, Shen, 2005]. These foundations have motivated context-sensitive adaptations, such as asymmetric scoring rules [Hand and Anagnostopoulos, 2014], although adoption remains uneven (see Appendix A for details on the difficulty of picking Beta parameters).

Recent efforts in calibration [Platt, 1999, Guo et al., 2017], fairness-aware evaluation [Hebert-Johnson et al., 2018], and robustness under label shift [Moreno-Torres et al., 2012, Muandet et al., 2013] all aim to bridge the gap between formal guarantees and practical constraints. Meanwhile, AUC-based metrics continue to face scrutiny over interpretability and subgroup fairness [Kallus and Zhou, 2019], prompting exploration of cost-sensitive methods [Elkan, 2001] and visualization-based diagnostics [Murphy, 1977, Ehm et al., 2016].

Frequent suggestions are made to measure both AUC-ROC and Expected Calibration Error (ECE) Pakdaman Naeini et al. [2015], despite known critiques of ECE [Vaicenavicius et al., 2019, Widmann et al., 2019, Nixon et al., 2020, Futami and Fujisawa, 2025, Roelofs et al., 2022, Ferrer, 2022] and the lack of a principled way to combine the two scores.

Our work contributes to this evolving landscape by proposing an evaluation framework that preserves the rigor of proper scoring rule theory while introducing design flexibility to reflect bounded cost asymmetries and decision-relevant dispersion. In doing so, we aim to retain the theoretical benefits of established metrics while addressing their practical limitations in contexts where calibration, subgroup validity, or threshold sensitivity are paramount.

## 2 Accuracy: Calibration without Label Shift Uncertainty

The most popular metric for evaluating binary classifiers is the simplest: accuracy.

**Definition 2.1** (Accuracy). Given a dataset $\mathcal{D}_{\boldsymbol{\pi_0}}$, a score function $s$, and a threshold $\tau$, the *accuracy* is defined as

$$\text{Accuracy}(\mathcal{D}_{\boldsymbol{\pi_0}}, s, \tau) = \frac{1}{|\mathcal{D}_{\boldsymbol{\pi_0}}|} \sum_{(x,y) \in \mathcal{D}_{\boldsymbol{\pi_0}}} \mathbf{1}_{\left(y = \kappa(s(x), \tau)\right)}$$

Accuracy considers the binarized score, discarding the real-valued information necessary for assessing calibration or uncertainty. It further assumes false positives and false negatives are equally costly. This is misaligned with most real-world decision problems where asymmetric stakes are the norm. Finally, the validity of the evaluation results presumes that the operational data-generating distribution matches the evaluation distribution, thereby ignoring the possibility of distribution shift. We describe existing extensions to address asymmetric costs, label shift, and calibration.

**Asymmetric Costs**   In most practical decision problems, false positives and false negatives carry asymmetric consequences. Several extensions of accuracy have been proposed to account for this asymmetry. Two commonly used variants are net benefit and weighted accuracy. The notion of net benefit that we use originates from decision curve analysis (DCA) [Vickers and Elkin, 2006] but is similar in structure to earlier formulations [Murphy, 1966]. We use a variation of net benefit parameterized by the benefit of true negatives rather than the costs of false positives in order to be more directly comparable to accuracy.

**Definition 2.2** (Net Benefit). Given a dataset $\mathcal{D}_{\pi_0}$, a score function $s$, a threshold $\tau$, and cost parameter $c \in (0,1)$, the *net benefit* is defined as

$$\text{Net Benefit}(\mathcal{D}_{\pi_0}, s, \tau, c) = \frac{1}{|\mathcal{D}_{\pi_0}|} \sum_{(x,y) \in \mathcal{D}_{\pi_0}} \left(\frac{c}{1-c}\right)^{1-y} \mathbf{1}_{\left(y = \kappa(s(x), \tau)\right)}$$

However, the procedure by which the threshold should be set given what we now know about cost ratio in deployment remains unclear: namely, it will often not match the cost ratio used in training. If a score function $s(x)$ is well-calibrated, we can reliably threshold it to optimize binary decisions under any cost asymmetry. Specifically, the optimal threshold $\tau$ satisfies $P(Y = 1|s(x) = \tau) = \frac{V(0,1)-V(0,0)}{V(0,1)-V(0,0)+V(1,1)-V(1,0)}$, where $V$ is the value function. See Appendix B.1 for details. Net benefit has the advantage that the interpretation of true positives remains consistent with standard accuracy. That is, true positives are rewarded uniformly regardless of the cost ratio, while false positives are penalized according to the cost ratio determined by $c$. Another popular metric is *weighted accuracy*, which corresponds to what Murphy [1966] called relative utility, and is normalized so that a perfect classifier achieves a score of 1 regardless of class balance. We provide a definition in Appendix C. While net benefit and weighted accuracy are both widely used, both inherit critical limitations from the basic accuracy framework: they binarize the score, thereby discarding information about uncertainty and calibration, and they assume a fixed data-generating distribution, thereby failing to account for distribution shift.

## 2.1 Label Shift

To model deployment scenarios, we adopt a causal perspective. Under the label shift assumption, represented by the causal structure $\mathcal{D}_\pi \rightarrow Y \rightarrow X$, the conditional distribution $P(X \mid Y, \mathcal{D}_\pi) = P(X \mid Y)$ remains invariant across domains. This assumption holds in many clinical contexts, where observed features ($X$) reflect underlying conditions ($Y$) whose prevalence varies across populations ($\mathcal{D}_\pi$). We focus on this structure because it aligns with the intuition of identifying latent diagnostic classes and enables robust correction methods for distribution shift. In contrast, the alternative causal structure $\mathcal{D}_\pi \rightarrow X \rightarrow Y$ is often associated with time-to-event outcomes, requiring distinct modeling strategies such as survival analysis.

The $\mathcal{D}_\pi \rightarrow Y \rightarrow X$ structure permits importance sampling to estimate deployment-time expectations. However, because prediction is performed via $P(Y \mid X)$, we must also adjust the posterior using Bayes' rule to account for class prevalence changes [Meehl and Rosen, 1955, Heckman, 1979, Saerens et al., 2002]. This yields the adjusted posterior:

$$P(Y = 1|s(x), \mathcal{D}_\pi) = P(Y = 1|s(x), \mathcal{D}_{\pi_0}) \otimes (1 - \pi_0) \otimes \pi.$$

We can then predict the net benefit attainable in the deployment environment if the score is adjusted using this formula. We define $s_{1/2}(x) \triangleq 1 - \pi_0 \otimes s(x)$, and denote the adjusted binary classifier by $\kappa(\pi \otimes s_{1/2}(x), \tau)$. The full derivation is provided in Appendix B.3.

**Definition 2.3** (Prior-Adjusted Maximum Net Benefit). Given an empirical dataset $\mathcal{D}_{\pi_0}$, a score function $s$, a threshold $\tau$, the deployment distribution $\mathcal{D}_\pi$, and the cost ratio parameter $c \in (0, 1)$, the *prior-adjusted maximum net benefit* is defined as,

$$\text{PAMNB}(\mathcal{D}_\pi, s, \tau, c) = \frac{1}{|\mathcal{D}_{\pi_0}|} \sum_{x, y \in \mathcal{D}_{\pi_0}} \left(\frac{\pi}{\pi_0}\right)^y \left(\frac{c}{1-c}\frac{1-\pi}{1-\pi_0}\right)^{1-y} \mathbf{1}_{\left(y = \kappa(\pi \otimes s_{1/2}(x), c)\right)}.$$

The above expression generalizes several familiar metrics. When $c = 1/2$ it reduces to the *prior-adjusted maximum accuracy (PAMA)* (see Definition C.3), recovering the standard decision-optimal accuracy under label shift. For arbitrary $c$, it provides a cost-sensitive extension that supports domain adaptation under both class imbalance and asymmetric decision costs. While this approach can fully adjust for any fixed label shift, it still relies on binarized scores and thus cannot capture uncertainty over the true deployment prevalence.

## 2.2 Calibration

Beyond adapting to shifts in population prevalence, another crucial aspect of evaluating probabilistic model outputs, particularly in clinical decision-making, is *calibration*. Unlike threshold-based

decision-making, which focuses on classification accuracy, the goal of calibration is to ensure that predicted probabilities match observed frequencies: that is, $P(Y = 1|s(x)) = s(x)$. This perspective, well-established in the weather forecasting literature, prioritizes reporting reliable probabilities over optimizing decisions directly. However, calibration alone does not guarantee utility.

Consider the ACC/AHA guidelines for primary prevention of cardiovascular disease. The guidelines recommend prescribing statins based on 10-year cardiovascular disease risk, thresholded at 2.5%, 5%, 7.5%, and 20% [Arnett et al., 2019]. A calibration-only approach can be pathological: a model that predicts the population base risk (around 10%) for everyone would receive an excellent score from calibration-only metrics like ECE [Ferrer and Ramos, 2024], but it would be clinically unhelpful because it recommends a light dose of statins for every patient. It is easiest to understand optimal thresholding if a classifier has everywhere the property that $P(Y = 1|s(x) = c) = c$. However, what is practically necessary for a classifier to support good thresholding at $c$ is that $P(Y = 1|s(x) > \tau) > c$ and $P(Y = 0|s(x) < \tau) < c$, even if it is miscalibrated elsewhere. In the ACC/AHA example, in the absence of deployment shift, we would need this property to hold at 2.5%, 5%, 7.5%, and 20%; it is not necessary for it to hold elsewhere.

Uncertain label shift is more complex and motivates a need for a broader sense of calibration. If we can bound the possible class balances, we will need the model to be calibrated within the whole corresponding range. A score function that is well-calibrated only in this narrow region can, however, still support robust, cost-sensitive classification. This suggests a more nuanced perspective: rather than enforcing global calibration, it may suffice to ensure calibration (by measuring miscalibration) within a threshold band. Part of the contribution of this paper is to formalize and operationalize this idea.

## 3 AUC-ROC: Label Shift Uncertainty without Calibration

The most common approach to integrate over a range of operating conditions is to use AUC-ROC instead of accuracy. AUC-ROC is an ordinal metric that is still based on thresholding, but it discards information about the magnitudes of those thresholds and evaluates performance solely based on the relative ordering between positive and negative examples.

**Definition 3.1** (AUC-ROC). Let $s : \mathcal{X} \to \mathbb{R}$ be a scoring function on $\mathcal{D}_{\pi_0}$, the training data distribution. The AUC-ROC is given by

$$\text{AUC-ROC}(\mathcal{D}_{\pi_0}, s) \triangleq \sum_{(x,y) \in \mathcal{D}_{\pi_0}} \frac{1}{|\mathcal{D}_{\pi_0}|} \frac{1-y}{1-\pi_0} \sum_{(x',y') \in \mathcal{D}_{\pi_0}} \frac{1}{|\mathcal{D}_{\pi_0}|} \frac{y'}{\pi_0} \left[ \mathbf{1}_{(s(x')>s(x))} + \tfrac{1}{2} \mathbf{1}_{(s(x')=s(x))} \right]$$

At first glance, this formulation poses a challenge for decision-theoretic interpretation. Specifically, it is not *a priori* clear how to interpret AUC-ROC within a framework where the metric corresponds to expected utility or decision quality under a specified loss function and distributional assumption. AUC-ROC resists this interpretation because it is invariant to monotonic transformations of the score function and, therefore, is indifferent to the calibration or absolute values of the scores, which are central to threshold-based decision-making. On the other hand, AUC-ROC does capture something that accuracy fails to: it aggregates performance across the full range of the score distribution, effectively summing a population-level statistic over levels of the score.

There are numerous ways to interpret AUC-ROC, at least a dozen of which are enumerated in Appendix G. We nevertheless offer a new formulation, whose proof is in Theorem H.5, that sheds particular light on its relationship to label shift, i.e. when the marginal distribution over labels differs between training and deployment.

**Theorem 3.2** (AUC-ROC as Accuracy Averaged Across Label Shift). *Let $s$ be a scoring function that is calibrated on the evaluation distribution $\mathcal{D}_{\pi_0}$. Then:*

$$\text{AUC-ROC}(s) = \frac{1}{2} \mathbb{E}_{t \sim s[\mathcal{D}_{1/2}]}[\text{PAMNB}(\mathcal{D}_{1-t}, s, 1/2, 1/2)]$$

*where $\mathcal{D}_{1/2}$ denotes a balanced reweighting of the dataset (i.e., class prior $\pi = 1/2$), and $s[\mathcal{D}_{1/2}]$ denotes the distribution of model scores over this reweighted set.*

This perspective reveals that AUC-ROC can be viewed as averaging thresholded accuracy across a distribution of class prevalences, albeit one that is induced implicitly by the score distribution of

the model itself. This provides a limited form of robustness to label shift in contrast to metrics like accuracy which are typically evaluated at a fixed class balance. Yet this same interpretation also exposes several critical limitations.

First, AUC-ROC entirely disregards calibration. By evaluating only the ordering of scores, it fails to assess whether predicted probabilities are well-aligned with empirical outcomes, which is a property essential in many domains where decisions depend on absolute risk levels. This issue is shared by other ranking metrics such as AUC-PR and *net discounted cumulative gain*, which similarly ignore score magnitudes. A frequent suggestion made in response is to report ranking metrics together with binned calibration metrics, but there is in general no principled way to combine the two scores to make a decision. As shown in Table 1, real clinical data can produce cases where AUC-ROC and ECE together fail to indicate which subgroup performs better, underscoring the need for unified evaluation criteria.

Second, although AUC-ROC evaluates calibrated scores for their performance across varying class balances, the distribution over these prevalences is not user-specified or interpretable. It is instead a byproduct of the model's score distribution on a hypothetical balanced dataset. Consequently, the underlying population over which AUC-ROC aggregates accuracy differs across models, making metric comparisons across models trained on the same data unreliable.

Finally, AUC-ROC does not allow the independent specification of label shift and asymmetric error costs. Although we can interpret varying prevalences as including varying cost ratios through the relationship $\pi' = 1 - c \otimes \pi$ [Hernandez-Orallo et al., 2011], doing so entangles cost asymmetry with shifts in class balance.

In summary, AUC-ROC offers a partial advantage over accuracy by aggregating across class balances, but its benefits are offset by its insensitivity to calibration, its implicit and model-dependent averaging distribution, and its inability to account for cost asymmetry. While it captures ranking performance, it fails to reflect key aspects of real-world decision quality. The historical development of AUC-ROC provides important context. Ordinal metrics originated in fields like psychology, where class prevalences were fixed by design, and information retrieval, where results per page were fixed by the capacity constraint of the querying user regardless of quality. Their subsequent adoption in machine learning reflects a shift in evaluation priorities away from deployment evaluation and toward the abstract comparison of new architectures and optimization techniques. In such a setting, ordinal metrics offer a convenient, threshold-free mode of comparison. However, such metrics are poorly aligned with the needs of real-world deployments, where thresholding, cost asymmetries, and calibration are often indispensable.

# 4 Log Score: Label Shift, Calibration and Cost Asymmetry

To evaluate the utility of a thresholded classifier under uncertain or varying class balance, it is critical to evaluate the calibration of its underlying score function across a range of label distributions. Calibration metrics are only meaningful insofar as they are expressed in units that reflect application-specific costs. In this context, cost asymmetry is not a minor adjustment but a first-order concern that must be explicitly accounted for. Moreover, a persistent challenge in real-world deployments is the difficulty of comparing the impact of miscalibration, measured in cost-aligned units, with the loss in performance attributable to poor sharpness or uncertainty in ranking [Vickers and Holland, 2021]. As we have seen, measuring both AUC-ROC and ECE separately does not solve this problem.

We build on the view that proper scoring rules can be interpreted as mixtures over a distribution $H$ of data distributions $\mathcal{D}_\pi$, where each scoring rule evaluates cost-weighted errors $V$ over the corresponding $\mathcal{D}_\pi$. Our approach does not have the ambiguity of combined cost / balance terms in Drummond and Holte [2006], nor does it require the problematic double integration over both cost and prevalence as suggested in Hand and Anagnostopoulos [2023]. Instead, we fix the cost ratio $c$ and integrate over the variability of data distributions captured by $H$, yielding tools that are computationally simpler and semantically interpretable.

## 4.1 Background

Accuracy can be generalized to account for asymmetric costs and label shift, but it assumes exact knowledge of a single value of class balance in deployment. The log score (or cross entropy) can,

owing to the Schervish representation, be viewed as an average of accuracy over the full range of possible class balances, from 0 to 1. Unfortunately, as Assel et al. [2017] point out, averaging over so wide a range is not specific enough to be clinically useful. We need, therefore, to find a way to narrow that range. When there are only a couple deployment settings, we can take a discrete average over performance in each. But as uncertainty grows we need a simpler and more reproducible continuous approach to narrowing the distribution.

## 4.2 Clipped Cross Entropy

The core contribution of this paper is to propose: (1) a simple way to characterize uncertainty in label shift using lower and upper bounds on class balance, (2) a straightforward means to average accuracy over that range, and (3) a natural extension to two standard approaches for handling asymmetric costs. As previously mentioned, there exists a duality between measures of calibration and mixtures of accuracy measures across different prevalences. This result is somewhat unintuitive; see Appendix D for a derivation that sheds more light. We extend this result to average over only a specific subinterval of prevalences. Our new contribution demonstrates how this can also be applied when costs are asymmetric. These formulas enable straightforward calculation of the average cost-sensitive performance of a classifier over a specified range of prevalences. A major practical benefit of this approach is that it is based on a pointwise calculation of loss, so confidence intervals can be trivially bootstrapped by resampling calculated losses, and each new draw only requires a weighted sum of these losses.

**Bounding the Class Balance**    We begin by specifying a lower bound $a$ and an upper bound $b$ on the class balance. These bounds can be obtained in direct consultation with domain experts, through surveys, or utilizing previous estimates. Consider the range of syphilis prevalence in US states: between about 7 in 10,000 (South Dakota) and 7 in 1,000,000 (Vermont) [U.S. Centers for Disease Control and Prevention, 2024]. Though two orders of magnitude wide, this interval still substantially constrains the range of possible prevalences.

Intuitively, practitioners might expect about half of the interval should lie in the lower order of magnitude, and half in the higher. However, if prevalence is uniformly distributed between South Dakota and Vermont, the midpoint is as high as 3 in 10,000 (Mississippi). For a midpoint around 7 in 100,000 (Connecticut), we instead want the log odds of the prevalence to be uniformly distributed. We represent this log odds transformation by saying that $\sigma^{-1}(\pi) \sim \text{Uniform}(\sigma^{-1}(a), \sigma^{-1}(b))$.

**Theorem 4.1** (Bounded DCA Log Score). *Let $s(x) \in [0, 1]$ be a score function and $c \in (0, 1)$ be a cost parameter defining a decision threshold and $\gamma \triangleq (2(1 - c)^{-1})/(\sigma^{-1}(b) - \sigma^{-1}(a))$ be a normalizing constant. Then, the expected net benefit over a logit-uniform prior on prevalence satisfies*

$$\mathbb{E}[\text{PAMNB}(\mathcal{D}_{\boldsymbol{\pi}}, s, \tau, c)] = \gamma(\mathbb{E}[\log|1 - y - \underset{[1-b,1-a]}{\text{clip}}((1 - c) \otimes s_{1/2}(x))| - \log|1 - y - \underset{[1-b,1-a]}{\text{clip}}(1 - y)|])$$

*where the expectation on the left-hand side is with respect to $\sigma^{-1}(\pi) \sim \text{Uniform}(\sigma^{-1}(a), \sigma^{-1}(b))$, and on the right-hand side is with respect to $(x, y) \sim \mathcal{D}_{(1-c)}$.*

*Proof.* See Theorem F.1.                                                                                         □

By clipping the score, this formula is able to produce a closed form expression for the average net benefit over a range of class balances. We focus on net benefit, because it is scaled the same way at different class balances, so it makes sense to add and average this quantity. However, see Theorem E.2 for a related derivation that holds for weighted accuracy as well; this is effectively a rescaled version of net benefit for which the maximum possible score of a perfect classifier is always 1, regardless of the class balance.

## 4.3 Calibration, Label Shift and Asymmetric costs.

The interpretation of our newly introduced DCA log score as a mixture of cost-weighted accuracy is clear: the units are, in all cases, units of true positives. The Schervish representation gives us a simple way to describe what this is doing as a calibration metric as well: it is calibrating the model only over a particular bounded range of scores that is relevant to decisions and weighting scores in that range uniformly in log odds space. Moreover, our approach is flexible enough that we can generalize

beyond accuracy (a well known result) to cost-sensitive metrics, as we demonstrate with DCA Log Score and Weighted Accuracy Log Score (See Appendix E for details). This, then, is a measure of miscalibration that can be compared directly for its effects in the world.

Moreover, Gneiting and Raftery [2007] showed that the unconstrained log score can be decomposed linearly into components of calibration and sharpness. As Vickers and Holland [2021] has argued, the need to weigh failures of calibration against failures of sharpness is a perpetual problem in the deployment of machine learning models in a medical context; one that measuring both AUC-ROC and ECE separately cannot resolve.

The Bounded DCA log score can easily be decomposed into a sum of miscalibration loss and sharpness, simply by recalibrating a version of the model on the test set (denoted $s^*$). The sharpness of the original and recalibrated models is the same, and is measured by:

$$\mathbb{E}[\text{PAMNB}_{\text{sharpness}}(\mathcal{D}, s, \tau, c)] = \mathbb{E}[\text{PAMNB}(\mathcal{D}, s^*, \tau, c)].$$

The miscalibration loss is the difference between the original and recalibrated models given by:

$$\mathbb{E}[\text{PAMNB}_{\text{miscalibration}}(\mathcal{D}, s, \tau, c)] = \mathbb{E}[\text{PAMNB}(\mathcal{D}, s, \tau, c)] - \mathbb{E}[\text{PAMNB}(\mathcal{D}, s^*, \tau, c)].$$

See Appendix I.3 for an example with real clinical data. Although the sharpness difference measured by clipped cross-entropy is weighted differently than that measured by AUC-ROC, the quantities are typically correlated.

## 5 Empirical Example: Subgroup Decomposition for eICU Mortality

We illustrate our framework by analyzing subgroup performance of the APACHE IV mortality prediction model in the publicly available eICU database [Goldberger et al., 2000, Pollard et al., 2018, 2019, Johnson et al., 2021]. The dataset contains electronic health records from ICUs across the US, including vital signs, laboratory values, diagnoses, and pre-computed APACHE severity scores [Zimmerman et al., 2006]. We use the model's existing in-hospital mortality predictions rather than retraining, enabling fully reproducible results with minimal computation. Further details on patient demographics are provided in Table 12. Our analysis focuses on two subgroup comparisons, by race and by sex, to demonstrate how the proposed metrics enable interpretable decomposition of model performance. Specifically, they (1) permit principled trade-offs between calibration and discrimination, (2) remain robust to prevalence differences, and (3) support restriction to clinically meaningful prevalence ranges.

**AUC-ROC vs Bounded DCA Log Score** Both conventional and proposed calibration measures indicate that APACHE IV is better calibrated for African-American patients but more discriminative for Caucasian patients (Table 1). Unlike AUC-ROC and ECE, whose combination is ill-defined, the Bounded DCA Log Score integrates calibration and discrimination on a common probabilistic scale, yielding a single interpretable measure of overall decision quality. These results can be

| Metric | Caucasian | African-American |
|---|---|---|
| ↑ Bounded DCA Log Score (calibration-only) | **0.999** | 0.927 |
| ↓ ECE | **0.829** | 0.889 |
| ↑ Bounded DCA Log Score (discrimination-only) | 0.957 | **0.973** |
| ↑ AUC-ROC | 0.868 | **0.907** |
| ↑ Bounded DCA Log Score | **0.956** | 0.900 |
| ↓ ECE & ↑ AUC-ROC | ? | ? |

Table 1: Calibration and Discrimination for Caucasian and African-American Patients

decomposed into calibration and sharpness components (see Appendix I.3 for visualization). The decomposition confirms that most of the performance gap arises from calibration differences rather than discrimination.

**Accuracy vs Bounded DCA Log Score** We evaluate subgroup performance across prevalences by plotting the accuracy curve and summarizing it with the Bounded DCA Log Score (Figure 1).

Accuracy at any given prevalence (solid lines) is consistently higher for Caucasian patients than for African-American patients, indicating better discrimination across the full prevalence range. The Bounded DCA Log Score (horizontal bars) averages these accuracy curves over prevalences, providing a single scalar that correctly preserves the same ranking. At the actual label prevalence (circles), however, accuracy is higher for African-American patients. That is, the apparent advantage reverses when comparing at a single empirical prevalence rather than across the full range. This difference highlights how the gap in accuracy at a fixed prevalence reflects **label distribution**, whereas the gap in average accuracy reflects **model mechanism**. See Appendix I.2 for more details.

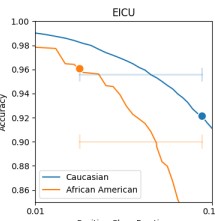

(a) Accuracy vs Prevalence curves and Bounded DCA Log Score (bars)

| Metric | Caucasian | African-American |
|---|---|---|
| ↑ Accuracy Curve@1% | **0.990** | 0.978 |
| ↑ Accuracy Curve@5% | **0.953** | 0.896 |
| ↑ Accuracy Curve@10% | **0.912** | 0.793 |
| ↑ Bounded DCA Log Score | **0.956** | 0.900 |
| ↑ Accuracy | 0.922 | **0.961** |

(b) Accuracy at varying prevalences.

Figure 1: Comparison of subgroup performance using accuracy and the Bounded DCA Log Score.

**Cross Entropy vs Bounded DCA Log Score** At low prevalences (those most clinically relevant) APACHE IV achieves higher accuracy for female than for male patients (Table 2a). At balanced or high prevalences, accuracy slightly favors males, but these regions contribute disproportionately to cross-entropy, which therefore ranks male performance higher overall. By emphasizing plausible prevalence ranges, the Bounded DCA Log Score (Table 2b) corrects this bias, ranking female patients higher. The corresponding accuracy–prevalence curves appear in Appendix I.4.

| Metric | Male | Female |
|---|---|---|
| ↑ Accuracy@1% | 0.989 | 0.990 |
| ↑ Accuracy@5% | 0.948 | 0.954 |
| ↑ Accuracy@25% | 0.818 | 0.814 |
| ↑ Accuracy@75% | 0.846 | 0.832 |
| ↑ Accuracy@95% | 0.953 | 0.953 |
| ↑ Accuracy@99% | 0.990 | 0.990 |

(a) Accuracy across prevalences.

| Metric | Male | Female |
|---|---|---|
| ↓ Brier | 0.062 | 0.062 |
| ↓ Cross-Entropy / NLL | 0.214 | 0.218 |
| ↑ Bounded DCA Log Score | 0.917 | 0.926 |

(b) Scoring rules comparison.

Figure 2: Comparison of APACHE IV performance for male and female patients using accuracy across prevalences (a) and proper scoring rules (b).

**Summary** These analyses show that ranking metrics such as AUC-ROC fail to capture overall usefulness on their own, and cannot be meaningfully combined with separate calibration metrics. Accuracy is similarly limited by its mixture of prevalence differences and model mechanism differences. Meanwhile, unbounded scoring rules such as cross-entropy and Brier score put too much weight on clinically unrealistic prevalences, echoing earlier critiques Assel et al. [2017]. In contrast, the Bounded DCA Log Score provides an interpretable, unified evaluation across subgroups and prevalence ranges. Confidence intervals are reported in Appendix I.5; given that mortality prevalence and African-American representation are both about 10% in the public subset, these results should be viewed as illustrative rather than conclusive.

## 6 Discussion

The prevailing paradigm for evaluating medical ML decision-support systems often misaligns with evidence-based medicine and beneficence by overlooking real-world cost structures, disease prevalences, and calibration nuances. While label shift correction and cost-sensitive learning are well

developed fields, evaluation methods to adapt to them are less widespread; for example, a recent review of 173 papers using cost-sensitive learning in medicine found less than 10% of papers actually used existing cost-sensitive metrics [Araf et al., 2024]. Evaluating sharpness and calibration separately, with incommensurable metrics, remains the standard practice. We wish to emphasize that, to our knowledge, this is the first framework to combine these three dimensions into a unified, practically deployable metric, directly addressing the persistent challenge in clinical ML of actionable model selection under label shift and asymmetric costs.

**Implications for Practice**   The key insight is that *eliciting decision context precedes evaluation*. Assessing model utility requires understanding which tradeoffs matter and which populations will be served. At the same time, practitioners need tools to guide this dialogue. Visualizing accuracy across prevalences (Figure 1a) provides a concrete basis for conversation: "Here is how the model performs at 2% versus 10 % prevalence — what range matters for our setting?" Once relative costs and plausible prevalence bounds are elicited, our framework converts these into a cost-weighted average performance score, identifying the model expected to be most useful across deployments. This bridges the gap between theoretical desiderata and the operational realities of medical practice.

**Limitations and Future Work**   While our framework offers a unified and interpretable approach to model evaluation, several theoretical and practical challenges remain.

*Cost Uncertainty.* Our analysis assumes known cost ratios. Naively integrating DCA log score over cost ratios introduces dilogarithmic expressions that are analytically opaque and limit practical interpretability and scalability. Future work could explore tractable approximations or surrogate objectives, such as piecewise-linear or entropy-regularized alternatives, that preserve sensitivity to cost uncertainty while enabling smoother optimization and interpretive clarity.

*Sampling Variability under Label Shift.* In settings with symmetric misclassification costs, bootstrap resampling or binomial confidence intervals suffice for uncertainty estimation. However, under asymmetric costs, especially with population label shift, the evaluation metrics become sensitive to multinomial fluctuations in both the score distribution and the cost-weighted outcome prevalence. This introduces more variance and potentially estimation bias. Quantifying and stabilizing this variability, possibly through Bayesian poststratification or domain-adaptive resampling methods, remains a significant challenge.

*Adaptive Base Rate Estimation.* Our framework presumes class prevalence is uncertain during model evaluation but fully known once the model is used in deployment. In practice, these rates may be uncertain or drift over time due to changing patient populations, care protocols, or screening policies. Jointly estimating prevalence and calibrating predictions in such regimes introduces second-order uncertainty. Future work could explore integrated approaches that dynamically re-estimate base rates (e.g. via semi-supervised learning or risk-set modeling) while propagating this uncertainty into threshold selection and cost evaluation.

*Asymmetric Cost Parameterization.* We adopt a general framework for asymmetric cost modeling, but the semantics of varying cost ratios remain under-theorized. Distinct parameterizations, such as utility ratios, slope-based decision thresholds, or expected loss curves, embed different assumptions about clinical values and tradeoffs. A systematic comparative study of these parameterizations could yield more robust and practitioner-interpretable guidelines for aligning model evaluation with domain-specific norms.

**Conclusion**   By combining decision-theoretic tools, causal framing, and clinically grounded metrics of calibration, this work moves toward evaluation methodologies that are both conceptually principled and actionable in real-world medical settings. Continued advances will require deeper integration of uncertainty quantification, model adaptivity, and domain-informed cost modeling.

# 7   Societal Impact

This paper is concerned with evaluating binary classifiers with due consideration of the decision context. Any incremental improvement to the process of machine learning deployment can aid the malicious actor as well as the righteous. However, we believe that the harm caused by ML models that are poorly matched to their deployment conditions is the primary area of concern here.

## Acknowledgements

GF and AW are generously supported by the MIT Jameel Clinic in collaboration with Massachusetts General Brigham Hospital and Simons Foundation Collaboration #733792 on Algorithmic Fairness.

A.H.S. acknowledges support from the National Science Foundation Graduate Research Fellowship Program under Grant #1938052. Any opinions, findings, and conclusions or recommendations expressed in this material are those of the authors and do not necessarily reflect the views of the National Science Foundation.

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

# A    Related Work

Recent literature emphasizes that scoring rules in AI should meaningfully reflect the objectives of deployment contexts, rather than relying on standard metrics that can lead to suboptimal or misleading outcomes Cruz Rivera et al. [2020], Liu et al. [2020].

**Decision Theory.**    Decision theory has roots in gambling and actuarial sciences but was formally structured by foundational works such as Ramsey [1926] and de Finetti [1937, 1992]. Within medical decision-making, a prominent recent strand has been Decision Curve Analysis (DCA), developed by Vickers and Elkin [2006], Vickers et al. [2019], a decision-theoretic framework. However, DCA has avoided measuring the area under the decision curve [Steyerberg and Vickers, 2008], preferring instead to eschew mathematical evaluation when neither classifier dominates. This body of work critically examines widely-used metrics such as the Area Under the Curve (AUC) [Vickers and Holland, 2021, Vickers and Woo, 2022] and the Brier score [Assel et al., 2017], questioning their clinical utility and advocating for metrics directly connected to decision-analytic value.

**Proper Scoring Rules.**    The literature on proper scoring rules began with Brier [1950], quickly enriched by contributions from Good [1952] and McCarthy [1956]. A critical advancement was the integral representation of proper scoring rules by Shuford et al. [1966], explicitly connecting scoring rules with decision-theoretic utility via Savage [1971]. This was followed by the comprehensive characterization provided by Schervish [1989], who demonstrated that strictly proper scoring rules can be represented as mixtures of cost-weighted errors. The formalism was further elucidated by Shen [2005] through the lens of Bregman divergences and by Gneiting and Raftery [2007] as convex combinations of cost-sensitive metrics. Independently, Hand [2009] rediscovered proper scoring rules, reframed as H-measures, in the context of mixtures over cost-weighted errors. He used this framing to show that the AUC-ROC of a calibrated classifier corresponds to a mixture of cost-weighted errors under a particular (and undesirable) distribution over cost ratios.

The idea of generalizing from cost to a cost proportion that depends also on class balance has been repeatedly independently proposed in the setting with known analytic distributions of scores Drummond and Holte [2006], Hernández-Orallo et al. [2012]. Hand and Anagnostopoulos [2023] introduced the idea of a double integral over cost and balance, but their work does not explore the semantics of the resulting joint distribution nor provide guidance on how the double integral should be computed.

These foundations have also motivated context-sensitive adaptations. Recent attempts have focused on obtaining a central estimate and fitting a Beta distribution around it Hand [2009], Hand and

Anagnostopoulos [2014], Zhu et al. [2024]. Unfortunately, the dispersion of a Beta distribution remains unintuitive to most medical (and perhaps even machine learning) practitioners. Indeed, Zhu et al. [2024] does not provide a procedure to set the pseudocount $\alpha + \beta$, and Hand and Anagnostopoulos [2014] do not quantify uncertainty, but suggest always using 3 as "a sensible default value".

**Calibration Techniques.** The Pool Adjacent Violators Algorithm (PAVA), introduced by Ayer et al. [1955], remains a foundational calibration technique, equivalent to computing the convex hull of the ROC curve [Fawcett and Niculescu-Mizil, 2007]. A distinct parametric calibration approach based on logistic regression was popularized by Platt [1999], subsequently refined for slope-only calibration by Guo et al. [2017]. An intercept-only version aligns closely with simple score adjustments [Saerens et al., 2002], while broader generalizations are explored in Kull et al. [2017]. More recently, the calibration literature shifted towards semi-supervised contexts, utilizing unlabeled data to enhance calibration quality [Lipton et al., 2018, Azizzadenesheli et al., 2019, Garg et al., 2020].

Despite extensive critiques, for example highlighting that the widely-adopted Expected Calibration Error (ECE) [Pakdaman Naeini et al., 2015] is not a proper scoring rule [Vaicenavicius et al., 2019, Widmann et al., 2019], and statistical issues with binned calibration metrics (of which ECE is an example) Nixon et al. [2020], Futami and Fujisawa [2025], Roelofs et al. [2022], Ferrer [2022] this metric remains popular in practice.

Parallel to predictive accuracy, calibration has also emerged as a fairness metric. This perspective, however, has become contentious since calibration was shown to be fundamentally incompatible with other fairness criteria [Kleinberg et al., 2017], spurring the development of "multicalibration" approaches that ensure calibration across numerous demographic subgroups [Hebert-Johnson et al., 2018].

**Label Shift.** Label shift techniques are a particularly useful hyponym of calibration techniques. While the concept of shifting class prevalences without altering the underlying conditional distribution of features is longstanding Meehl and Rosen [1955], Heckman [1974, 1979], Saerens et al. [2002], formal treatments and systematic causal characterizations arose from Moreno-Torres et al. [2012]. Earlier explorations of covariate shift [Sugiyama et al., 2008] motivated a broader field of research aimed at developing invariant representations robust to distribution shifts. These efforts encompass methods based on richer causal assumptions [Subbaswamy et al., 2018], invariant representation learning [Ben-David et al., 2010, Muandet et al., 2013], and distributionally robust optimization [Long et al., 2015, Ganin et al., 2016, Sagawa* et al., 2020, Duchi and Namkoong, 2021].

**AUC-ROC & AUC-PR.** The Receiver Operating Characteristic (ROC) curve emerged within signal detection theory [Peterson and Birdsall, 1953, Tanner et al., 1953], later becoming central in radiology and clinical diagnostics where the convention solidified around measuring performance via the Area Under the Curve (AUC) [Metz, 1978, Hanley and McNeil, 1982]. Use of AUC to aggregate over multiple thresholds was explored by Spackman [1989], Bradley [1997], and Huang and Ling [2005], with subsequent critiques noting widespread interpretability issues [Carrington et al., 2023]. Hand [2009] showed how the AUC of calibrated classifiers relates to average accuracy across thresholds, while Hernández-Orallo et al. [2012] described alternative interpretations via uniform distributions of predicted score, or uniform distributions of desired positive fractions (see Appendix G for more details).

Recently, there has been increased scrutiny of AUC-ROC, particularly regarding its lack of calibration and poor decomposability across subgroups [Kallus and Zhou, 2019]. Precision and Recall metrics originated in information retrieval, with Mean Average Precision (MAP) or the Area Under the Precision-Recall Curve (AUC-PR) formalized by Keen [1966, 1968]. While more recent trends in information retrieval have favored metrics such as Precision@K, Recall@K, and Discounted Cumulative Gain (DCG), Davis and Goadrich [2006] popularized AUC-PR for classifier evaluation, particularly in contexts with imbalanced data. Despite well-documented critiques—including that AUC-PR poorly estimates MAP [Blockeel et al., 2013] and lacks clear theoretical justification [McDermott et al., 2024], its use persists, particularly in medical and biomedical contexts.

**Cost-Sensitive Learning.** Cost-sensitive evaluation, historically formalized through Cost/Loss frameworks [Angstrom, 1922, Murphy, 1966], was independently introduced in clinical decision-making as early as Pauker and Kassirer [1975]. The modern foundation of cost-sensitive learning emerged prominently in the machine learning literature in the 1980s, notably via the seminal work on MetaCost by Domingos [1999] and the canonical overview by Elkan [2001]. Extending these

frameworks to multi-class settings poses challenges due to the quadratic complexity of pairwise misclassification costs.

**Visualization.** Visualization techniques to illustrate economic or decision-theoretic value as a function of decision thresholds date back to Thompson and Brier [1955], with subsequent development by Murphy [1977], Murphy and Winkler [1987], who linked visualizations explicitly to scoring rule theory. Later rediscoveries within machine learning were articulated by Adams and Hand [1999], and independently by Drummond and Holte [2000, 2006]. More recently, these visualizations were generalized to include uncalibrated models [Hernández-Orallo et al., 2011] and formally named Murphy Diagrams by Ehm et al. [2016], with further implementation guidance provided by Dimitriadis et al. [2024].

# B   Calibration

The weather forecasting literature focuses on what are known as strictly proper scoring rules: those metrics that have the property that a forecaster is correctly incentivized to report their actual beliefs about the probability of the event. At first glance this seems a bit distant from binary classifier evaluation. After all, action is generally binary; we really want the weather report to tell us whether to take an umbrella, not to give us 3 decimal places of precision on the long run frequency with which it would rain.

## B.1   Asymmetric Cost

However, a calibrated, thresholded binary classifier has an immensely useful property: we know how to change the threshold to trade off false positives for false negatives if the class balance or the cost ratio changes.

Let us define the value function of a particular true and predicted label pair as $V(y, \widehat{y})$.

| $V(y, \widehat{y})$ | $y{=}1$ (Syphilis) | $y{=}0$ (No Syphilis) |
|---|---|---|
| $\widehat{y}{=}1$ (Treat) | True Positive | False Positive |
| $\widehat{y}{=}0$ (Don't treat) | False Negative | True Negative |

Table 2: Value function

The optimality condition for choosing a threshold requires that the first derivative of the expected value be zero. This is equivalent to saying that the expected utility of assigning points exactly at the threshold to either class should be the same:

$$\underset{x,y:s(x)=\tau}{\mathbb{E}} V(y, 0) = \underset{x,y:s(x)=\tau}{\mathbb{E}} V(y, 1)$$

$$\implies P(y = 1|s(x) = \tau) = \frac{V(0, 1) - V(0, 0)}{(V(0, 1) - V(0, 0)) + (V(1, 1) - V(1, 0))}$$

As a result, we generally call the quantity on the right $c$ and use it to describe the asymmetry of the error costs.

Solving this for $\tau$ requires us to at least implicitly estimate $\tilde{s}(\tau) = P(y = 1|s(x) = \tau)$. If we add the constraint of monotonicity to $\tilde{s}(\tau)$, then this problem is known as isotonic regression, and there are well-known algorithms for solving it. Assuming the existence of a good estimator $\tilde{s}(\tau)$ for this quantity, then $\tilde{s}(s(x))$ is of course a calibrated estimator for $P(y = 1|x)$. Using the same classifier at varying cost asymmetries requires that the classifer be, at minimum, implicitly calibrated; isotonic regression is in fact how such classifiers are calibrated.

It is, of course, possible to develop an estimator calibrated only at $s(x) = c$; for any point higher or lower, ordinal comparison alone is enough to make a decision. A classifier optimized in this fashion may be wildly unreliable at other values of $s(x)$, and our calibration may simply give us two scores: higher than $c$ and lower than $c$. If so, the condition of calibration is almost trivially satisfied: it only requires a binary predicted label and statistics for how often the classifier is correct in either case (PPV and NPV).

## B.2 Label Shift

In the weather forecasting literature, it is explicitly understood that $\mathcal{D}_{\boldsymbol{\pi}} \to X \to Y$, which is to say that rather than today's atmospheric conditions being emanations of tomorrow's decision of whether to rain or not, the evolution on physical principles of today's conditions leads to tomorrow's weather. As such, the idea of label shift is incoherent. The study of changes in classifier performance when $P(y)$ changes in this setting is known as covariate shift; this is out of scope for this paper.

The machine learning evaluation literature does acknowledge links between label shift and calibration. However, the setting is more abstract, with CDFs taken as given, and varying thresholds interpreted as a response to varying class balances without a clear enumeration of assumptions.

### B.2.1 Notation

**Lemma B.1** (Importance Sampling as $\ell_1$ distance). *Consider the standard importance sampling weights to move from the training ($\pi_0$) to the deployment ($\pi$) distribution. Label shift always holds when reweighting data by class because after we stratify by class, we do not change the distribution within the class.*

$$
\begin{aligned}
W(\pi_0 \to \pi; y') &\triangleq \frac{\mathbb{P}_\pi(x, y = y')}{\mathbb{P}_{\pi_0}(x, y = y')} \\[2mm]
&= \underbrace{\frac{\mathbb{P}_\pi(x|y = y')}{\mathbb{P}_{\pi_0}(x|y = y')}}_{\textit{one, by label shift}} \frac{\mathbb{P}_\pi(y = y')}{\mathbb{P}_{\pi_0}(y = y')} \\[4mm]
&= \frac{^1/_2}{\mathbb{P}_{\pi_0}(y = y')} \frac{\mathbb{P}_\pi(y = y')}{^1/_2} \\[4mm]
&= W(\pi_0 \to {}^1/_2; y') \cdot 2\mathbb{P}_\pi(y = y') \\[4mm]
&= W(\pi_0 \to {}^1/_2; y') \begin{cases} 2(\pi - 0) & \textit{if } y' = 1 \\[2mm] 2(1 - \pi) & \textit{if } y' = 0 \end{cases} \\[4mm]
&= W(\pi_0 \to {}^1/_2; y') \; 2 \; |(1 - \pi) - y'|
\end{aligned}
$$

**Definition B.2** (Odds Multiplication).

$$
a \otimes b \triangleq \frac{ab}{ab + (1 - a)(1 - b)}
$$

**Proposition B.3** (Inverse).

$$
a \otimes b = c \quad \Longleftrightarrow \quad b = (1 - a) \otimes c
$$

**Proposition B.4** (Jacobian).

$$
a \otimes b \frac{da}{a(1 - a)} = a \otimes b \frac{d(a \otimes b)}{(a \otimes b)(1 - a \otimes b)}
$$

**Proposition B.5** (One minus distributes over odds multiplication).

$$
1 - (a \otimes b) = 1 - a \otimes 1 - b
$$

*Proof.*

$$
\begin{aligned}
\frac{ab}{ab + (1 - a)(1 - b)} + \frac{(1 - a)(1 - b)}{ab + (1 - a)(1 - b)} &= 1 \\
a \otimes b + \quad (1 - a) \otimes (1 - b) &= 1 \\
(1 - a) \otimes (1 - b) &= 1 - a \otimes b
\end{aligned}
$$

$\square$

**Proposition B.6** (Logit Odds Multiplication is Additive)**.**

$$\sigma^{-1}(a \otimes b) = \sigma^{-1}(a) + \sigma^{-1}(b)$$

*Proof.*

$$\log \frac{\frac{ab}{ab+(1-a)(1-b)}}{\frac{(1-a)(1-b)}{ab+(1-a)(1-b)}} = \log \frac{a}{1-a} + \log \frac{b}{1-b}$$

$\square$

**Proposition B.7** (Log Odds Interval Invariance)**.**

$$\sigma^{-1}(1 - c \otimes b) - \sigma^{-1}(1 - c \otimes a) = \sigma^{-1}(b) - \sigma^{-1}(a)$$

*Proof.*

$$
\begin{aligned}
&\sigma^{-1}(1 - c \otimes b) - \sigma^{-1}(1 - c \otimes a) \\
&= [\sigma^{-1}(1 - c) + \sigma^{-1}(b)] - [\sigma^{-1}(1 - c) + \sigma^{-1}(a)] \\
&= [\sigma^{-1}(1 - c) - \sigma^{-1}(1 - c)] + [\sigma^{-1}(b) - \sigma^{-1}(a)] \\
&= [\sigma^{-1}(b) - \sigma^{-1}(a)]
\end{aligned}
$$

$\square$

## B.3  Prior-Adjustment

With this notation, working with conditional probabilities is straightforward:

$$P(y = 1|s(x), \mathcal{D}_{\boldsymbol{\pi}})$$

$$= \frac{\overbrace{P(s(x)|y = 1, \mathcal{D}_{\boldsymbol{\pi}})}^{\text{collect this}} P(y = 1|\mathcal{D}_{\boldsymbol{\pi}})}{\underbrace{P(s(x)|y = 1, \mathcal{D}_{\boldsymbol{\pi}})}_{\text{and this}} P(y = 1|\mathcal{D}_{\boldsymbol{\pi}}) + \underbrace{P(s(x)|y = 0, \mathcal{D}_{\boldsymbol{\pi}})}_{\text{and this}} P(y = 0|\mathcal{D}_{\boldsymbol{\pi}})}$$

$$= \frac{P(s(x)|y = 1, \mathcal{D}_{\boldsymbol{\pi}})}{\underbrace{P(s(x)|y = 1, \mathcal{D}_{\boldsymbol{\pi}}) + P(s(x)|y = 0, \mathcal{D}_{\boldsymbol{\pi}})}_{\text{we can use label shift}}} \otimes P(y = 1|\mathcal{D}_{\boldsymbol{\pi}})$$

$$= \frac{P(s(x)|y = 1)}{\underbrace{P(s(x)|y = 1) + P(s(x)|y = 0)}_{\text{invariant to } \mathcal{D}_{\boldsymbol{\pi}}}} \otimes P(y = 1|\mathcal{D}_{\boldsymbol{\pi}})$$

$$P(y = 1|s(x), \mathcal{D}_{\boldsymbol{\pi}}) \otimes P(y = 0|\mathcal{D}_{\boldsymbol{\pi}}) = \underbrace{\frac{P(s(x)|y = 1)}{P(s(x)|y = 1) + P(s(x)|y = 0)}}_{\text{invariant to } \mathcal{D}_{\boldsymbol{\pi}}}$$

$$= P(y = 1|s(x), \mathcal{D}_{\boldsymbol{\pi_0}}) \otimes P(y = 0|\mathcal{D}_{\boldsymbol{\pi_0}})$$

$$P(y = 1|s(x), \mathcal{D}_{\boldsymbol{\pi}}) = P(y = 1|s(x), \mathcal{D}_{\boldsymbol{\pi_0}}) \otimes P(y = 0|\mathcal{D}_{\boldsymbol{\pi_0}}) \otimes P(y = 1|\mathcal{D}_{\boldsymbol{\pi}})$$

$$= P(y = 1|s(x), \mathcal{D}_{\boldsymbol{\pi_0}}) \otimes (1 - \pi_0) \otimes \pi$$

Here, the propagation of errors is straightforward: the log odds error will be the same size under both distributions, although of course the errors in probability space may be larger or smaller. Thus we can specify the best choice of prior-adjusted score:

## B.4  Prior-Adjusted, Cost-Weighted Threshold

Combining these two, we find that the best choice of decision threshold is

$$\pi \otimes (1 - \pi_0) \otimes s(x) \geq c$$

We will thus often refer to the induced optimal classifier $\kappa(\pi \otimes s_{1/2}(x), c)$.

## C   Derivation of Set-based Metrics

We start with the most popular family of evaluation methods, which are based on accuracy but include cost-sensitive generalizations. These only require a binary classifier, which we define as a function $\kappa(x) : \mathcal{X} \to \{0, 1\}$.

They differ along two axes: the way they factor in the cost of errors, and the way they factor in the class balance of the dataset.

|  | **Empirical** | **Balanced** | **Prior-Adjusted Maximum** |
|---|---|---|---|
| **Accuracy** | Accuracy | BA | PAMA |
| **Weighted Accuracy** | WA | BWA | PAMWA |
| **Net Benefit** | NB | BNB | PAMNB |

Table 3: Taxonomy of set-based evaluation metrics. Each row represents a different approach to handling error costs, and each column represents a different approach to handling class balance. Note that when balanced, the second and third rows are equivalent.

### C.1   Accuracy

**Definition C.1** (Accuracy). The accuracy of a thresholded binary classifier $\kappa(x, \tau)$ is given by:

$$\text{Accuracy}(\mathcal{D}_{\pi_0}, s, \tau) = \sum_{x,y \in \mathcal{D}_{\pi_0}} V_{1/2}(y, \kappa(s(x), \tau))$$

| $V_{1/2}(y, \widehat{y})$ | $y=1$ (Syphilis) | $y=0$ (No Syphilis) |
|---|---|---|
| $\widehat{y}=1$ (Treat) | 1 | 0 |
| $\widehat{y}=0$ (Don't treat) | 0 | 1 |

Table 4: Value function for Accuracy

This is impractically neutral with regard to cost in that $V(y, \widehat{y} = 1 - y)$ is not a function of y, which corresponds to the contingency table in Table 4. It is practical but neither neutral nor flexible with regard to distribution shift in the sense that it implicitly assumes: $H(\mathcal{D}_\pi) = \delta(\mathcal{D}_\pi = \mathcal{D}_{\pi_0})$.

The simplest way to make this more neutral is to evaluate on a balanced dataset, which we denote as $\mathcal{D}_{1/2}$. Mechanically, we can draw from this dataset using importance sampling, if we assume $\mathcal{D}_\pi \to Y \to X$ and therefore $P(X|Y, \mathcal{D}_\pi) = P(X|Y)$.

**Definition C.2** (Balanced Accuracy). The balanced accuracy of a thresholded binary classifier $\kappa(x, \tau)$ is given by:

$$\text{BA}(\mathcal{D}_{\pi_0}, s, \tau)$$
$$= \sum_{x,y \in \mathcal{D}_{1/2}} V_{1/2}(y, \kappa(s(x), \tau))$$
$$= \sum_{x,y \in \mathcal{D}_{\pi_0}} W(\pi_0 \to 1/2; y) V_{1/2}(y, \kappa(s(x), c))$$
$$= \sum_{x,y \in \mathcal{D}_{\pi_0}} V(y, \kappa(s(x), \tau))$$

| $V(y, \widehat{y})$ | $y=1$ (Syphilis) | $y=0$ (No Syphilis) |
|---|---|---|
| $\widehat{y}=1$ (Treat) | $\frac{1}{2\pi_0}$ | 0 |
| $\widehat{y}=0$ (Don't treat) | 0 | $\frac{1}{2(1-\pi_0)}$ |

Table 5: Value function for Balanced Accuracy

If more flexibility is desired, at the expense of neutrality, it is necessary to evaluate at an arbitrary class balance. Moreover, evaluating how well a classifier performs at one specific threshold is less useful than understanding how the best threshold performs at a specific class balance.

**Definition C.3** (Prior-Adjusted Maximum Accuracy). The prior-adjusted maximum accuracy given a scoring function $s$ and a threshold $\tau$ with a class balance $\pi$ is given by:

$$\text{PAMA}(\mathcal{D}_{\boldsymbol{\pi}}, s, \tau)$$

$$= \sum_{x,y\in\mathcal{D}_{\boldsymbol{\pi}}} V_{1/2}(y, \kappa(s(x), \tau))$$

$$= \sum_{x,y\in\mathcal{D}_{\boldsymbol{\pi_0}}} W(\pi \to {}^1\!/{}_2; y)V_{1/2}(y, \kappa(\pi \otimes s_{1/2}(x), \tau))$$

$$= \sum_{x,y\in\mathcal{D}_{\boldsymbol{\pi_0}}} V(y, \kappa(\pi \otimes s_{1/2}(x), \tau))$$

| $V(y, \widehat{y})$ | $y=1$ (Syphilis) | $y=0$ (No Syphilis) |
|---|---|---|
| $\widehat{y}=1$ (Treat) | $\frac{\pi}{\pi_0}$ | $0$ |
| $\widehat{y}=0$ (Don't treat) | $0$ | $\frac{1-\pi}{1-\pi_0}$ |

Table 6: Value function for Shifted Accuracy

## C.2 Weighted Accuracy

This problem is further complicated by the need to realistically confront asymmetric costs. Consider the syphilis testing case: unnecessary treatment is 10 to 100 times less costly than a missed detection. We will use 1/30 as a representative value for exposition, as the exact mechanics of syphilis testing are not central to this work.

First, we consider the balanced case, which is more mathematically tractable.

**Definition C.4** (Balanced Weighted Accuracy). The balanced weighted accuracy of a score function $s$ with a threshold $\tau$ is given by:

$$\text{BWA}(\mathcal{D}_{\boldsymbol{\pi_0}}, s, \tau, c)$$

$$= \sum_{x,y\in\mathcal{D}_{\boldsymbol{1/2}}} (1 - c)^y c^{1-y} V_{1/2}(y, \kappa(s(x), c))$$

$$= \sum_{x,y\in\mathcal{D}_{\boldsymbol{\pi_0}}} W(\pi_0 \to 1 - c; y)V_{1/2}(y, \kappa(s(x), c))$$

$$= \sum_{x,y\in\mathcal{D}_{\boldsymbol{\pi_0}}} V(y, \kappa(s(x), c))$$

| $V(y, \widehat{y})$ | $y=1$ (Syphilis) | $y=0$ (No Syphilis) |
|---|---|---|
| $\widehat{y}=1$ (Treat) | $\frac{1-c}{\pi_0}$ | $0$ |
| $\widehat{y}=0$ (Don't treat) | $0$ | $\frac{c}{1-\pi_0}$ |

Table 7: Value function for Balanced Weighted Accuracy

The minimum possible value of this expression is clearly 0 if $V(y, \widehat{y}) = 0$ for all $y$. The maximum is also clear:

$$\sum_{x,y\in\mathcal{D}_{\boldsymbol{\pi_0}}} W(\pi_0 \to {}^1\!/{}_2; y)W({}^1\!/{}_2 \to 1 - c; y)\mathbf{1}_{()} = (1 - c) + c = 1$$

The obvious combination of the two weighting terms is not correct, however.

$$\sum_{x,y\in\mathcal{D}_{\boldsymbol{\pi_0}}} W(\pi_0 \to {}^1\!/{}_2; y)W({}^1\!/{}_2 \to 1 - c; y)W({}^1\!/{}_2 \to \pi; y)\mathbf{1}_{()} = \pi(1 - c) + (1 - \pi)c \neq 1$$

The most intuitive approach involves rescaling the value of the true and false positives to be in the 1:30 ratio and then normalizing such that the maximum possible value remains 1 regardless of class balance. This is known as Weighted Accuracy. This procedure of normalizing the metric so that 0 is the worst possible value and 1 the best is generally known in the forecast evaluation literature as a skill score, but in the medical decisionmaking literature this particular metric is generally called Weighted Accuracy.

**Definition C.5** (Weighted Accuracy). The weighted accuracy of a thresholded binary classifier $\kappa(x, \tau)$ is given by:

$$\text{WA}(\boldsymbol{\mathcal{D}}_{\boldsymbol{\pi_0}}, s, \tau)$$

$$= \frac{\sum_{x,y \in \boldsymbol{\mathcal{D}}_{\boldsymbol{\pi_0}}} (1 - c)^y c^{1-y} V_{1/2}(y, \kappa(s(x), \tau))}{\sum_{x,y \in \boldsymbol{\mathcal{D}}_{\boldsymbol{\pi_0}}} (1 - c)^y c^{1-y} V_{1/2}(y, y)}$$

$$= \sum_{x,y \in \boldsymbol{\mathcal{D}}_{\boldsymbol{\pi_0}}} V(y, \kappa(s(x), \tau))$$

| $V(y, \widehat{y})$ | $y=1$ (Syphilis) | $y=0$ (No Syphilis) |
|---|---|---|
| $\widehat{y}=1$ (Treat) | $\frac{1-c}{(1-c)\pi_0 + c(1-\pi_0)}$ | 0 |
| $\widehat{y}=0$ (Don't treat) | 0 | $\frac{c}{(1-c)\pi_0 + c(1-\pi_0)}$ |

Table 8: Value function for Weighted Accuracy

## C.3 Net Benefit

However, this makes comparisons across different class balances less meaningful, since as the class balance varies, the normalizing factor changes. As a result, the effective value of a true positive changes. One common approach from the Decision Curve Analysis literature is instead to normalize the true positive to 1 and then rescale the false positive to keep the right ratio. The baseline in the DCA literature is to always predict the negative class, whereas the weighted accuracy literature uses a baseline of always predicting the wrong class. Since this is equivalent up to constants, we will modify the parameterization of Net Benefit to make it more directly comparable.

**Definition C.6** (Net Benefit). The net benefit of a scoring function $s$ with a threshold $\tau$ is given by:

$$\text{NB}(\boldsymbol{\mathcal{D}}_{\boldsymbol{\pi_0}}, s, \tau, c)$$

$$= \sum_{x,y \in \boldsymbol{\mathcal{D}}_{\boldsymbol{\pi_0}}} \left(\frac{c}{1-c}\right)^{1-y} V_{1/2}(y, \kappa(s(x), \tau))$$

$$= \sum_{x,y \in \boldsymbol{\mathcal{D}}_{\boldsymbol{\pi_0}}} V(y, \kappa(s(x), \tau))$$

| $V(y, \widehat{y})$ | $y=1$ (Syphilis) | $y=0$ (No Syphilis) |
|---|---|---|
| $\widehat{y}=1$ (Treat) | 1 | 0 |
| $\widehat{y}=0$ (Don't treat) | 0 | $\frac{c}{1-c}$ |

Table 9: Value function for Net Benefit

The disadvantage of this approach is that it's unintuitive that the net benefit of a perfect classifier is not reliably 1, and instead depends on the class balance. The advantage is that when comparing at different class balances, the value of a true positive and a true negative remain fixed, so measurements are directly compared on the same scale.

## C.4 Prior-Adjusted Maximum Cost-Weighted Metrics

We can combine the prior-adjusted maximum value approach with the cost-weighted metrics to get two new metrics that make sense to compare across different class balances.

**Definition C.7** (Prior-Adjusted Maximum Weighted Accuracy). The prior-adjusted maximum weighted accuracy given a scoring function $s$ and a threshold $\tau$ with a class balance $\pi$ is given by:

$\text{PAMWA}(\mathcal{D}_{\boldsymbol{\pi}}, s, \tau, c)$

$$= \frac{\sum\limits_{\mathcal{D}_{\boldsymbol{\pi}}} (1-c)^y c^{1-y} V_{1/2}(y, \widehat{y})}{\sum\limits_{\mathcal{D}_{\boldsymbol{\pi}}} (1-c)^y c^{1-y} V_{1/2}(y, y)}$$

$$= \frac{\sum\limits_{\mathcal{D}_{1/2}} [(1-c)\pi]^y [c(1-\pi)]^{1-y} V_{1/2}(y, \widehat{y})}{\sum\limits_{\mathcal{D}_{1/2}} [(1-c)\pi]^y [c(1-\pi)]^{1-y} V_{1/2}(y, y)}$$

$$= \frac{\sum\limits_{\mathcal{D}_{\pi_0}} [(1-c)\pi(1-\pi_0)]^y [c(1-\pi)\pi_0]^{1-y} V_{1/2}(y, \widehat{y})}{\sum\limits_{\mathcal{D}_{\pi_0}} [(1-c)\pi(1-\pi_0)]^y [c(1-\pi)\pi_0]^{1-y} V_{1/2}(y, y)}$$

$$= \sum\limits_{\mathcal{D}_{\pi_0}} V(y, \kappa(\pi \otimes s_{1/2}(x), \tau))$$

| $V$ | 1 | 0 |
|---|---|---|
| 1 | $\frac{1}{2\pi_0} \frac{(1-c)\pi}{(1-c)\pi + c(1-\pi)}$ | 0 |
| 0 | 0 | $\frac{1}{2(1-\pi_0)} \frac{c(1-\pi)}{(1-c)\pi + c(1-\pi)}$ |

Table 10: Value function for Prior-Adjusted Maximum Weighted Accuracy

**Proposition C.8** (PAMA Equivalence).

$$\text{PAMWA}(\mathcal{D}_{\boldsymbol{\pi}}, s, \tau, c) = \text{PAMA}(\mathcal{D}_{1-c \otimes \pi}, s, \tau)$$

*Proof.*

$\text{PAMWA}(\mathcal{D}_{\boldsymbol{\pi}}, s, \tau, c)$

$$= \frac{\sum\limits_{\mathcal{D}_{\boldsymbol{\pi}}} (1-c)^y c^{1-y} V_{1/2}(y, \widehat{y})}{\sum\limits_{\mathcal{D}_{\boldsymbol{\pi}}} (1-c)^y c^{1-y} V_{1/2}(y, y)}$$

$$= \frac{\sum\limits_{\mathcal{D}_{1/2}} [(1-c)\pi]^y [c(1-\pi)]^{1-y} V_{1/2}(y, \widehat{y})}{\sum\limits_{\mathcal{D}_{1/2}} [(1-c)\pi]^y [c(1-\pi)]^{1-y} V_{1/2}(y, y)}$$

$$= \frac{\sum\limits_{\mathcal{D}_{1-c \otimes \pi}} V_{1/2}(y, \widehat{y})}{\sum\limits_{\mathcal{D}_{1-c \otimes \pi}} V_{1/2}(y, y)}$$

$$= \text{PAMA}(\mathcal{D}_{1-c \otimes \pi}, s, \tau)$$

$\square$

We combine the ideas of Net Benefit (Definition C.6) adapted from the concepts used in the Decision Curve Analysis literature [Vickers and Elkin, 2006], with Prior-Adjusted Maximized metrics (Definition C.3) used in various settings, but originally adapted from base-rate adjustments in the psychometric literature [Meehl and Rosen, 1955] to define a new metric.

**Definition C.9** (Prior-Adjusted Maximum Net Benefit). The prior-adjusted maximum net benefit of a scoring function $s$ with a threshold $\tau$ with a class balance $\pi$ is given by:

$\text{PAMNB}(\mathcal{D}_{\boldsymbol{\pi}}, s, \tau, c)$

$$= \sum\limits_{\mathcal{D}_{\boldsymbol{\pi}}} (\frac{\pi}{\pi_0})^{1-y} (\frac{c}{1-c} \frac{1-\pi}{1-\pi_0})^{1-y} V_{1/2}(y, \widehat{y})$$

$$= \sum\limits_{\mathcal{D}_{\pi_0}} V(y, \kappa(\pi \otimes s_{1/2}(x), \tau))$$

| $V(y, \widehat{y})$ | $y=1$ (Syphilis) | $y=0$ (No Syphilis) |
|---|---|---|
| $\widehat{y}=1$ (Treat) | $\frac{\pi}{\pi_0}$ | 0 |
| $\widehat{y}=0$ (Don't treat) | 0 | $\frac{c}{1-c} \frac{1-\pi}{1-\pi_0}$ |

Table 11: Value function for Prior-Adjusted Maximum Net Benefit

We focus on the second because although the semantics of a single value are more confusing (since the perfect classifier is not normalized to 1), the values at different class balances are commensurable.

## C.5 Conclusion

If we are willing to accept the causal diagram $\mathcal{D}_{\pi} \rightarrow Y \rightarrow X$, then we have tools available in different parts of the literature to broaden $V(y, \kappa(x, \tau))$ to capture asymmetric costs and move from $H(\mathcal{D}_{\pi}) = \delta(\mathcal{D} = \mathcal{D}_{\pi_0})$ to $H(\mathcal{D}) = \delta(\mathcal{D} = \mathcal{D}_{\pi})$ for any given $\pi$.

# D Cost-sensitive Error Averaged Across Class Balances

A core obstacle to the use of proper scoring rules like the Brier Score to evaluate classifier performance under label shift in applied work is the ubiquity of asymmetric costs. Traditional derivations of the equivalence between Mean Squared Error and Average Accuracy over a range of class balances have taken the CDF of the positive and negative classes as given. These do not easily generalize to a world of asymmetric costs. This appendix develops a rigorous mathematical framework for analyzing cost-weighted binary classifier performance averaged across a range of class balances. The average of accuracy represents an integral over class balances of a sum over data of an $\ell^0$ ordering between the class balance and the model score on each data point. The broad outline of the approach is as follows:

$$\underbrace{\int}_{\ell^0(\pi)} \underbrace{\sum_{\mathcal{D}_{\pi}}} \ell^0(s(x), \pi, y) d\pi$$

$$= \underbrace{\int}_{\ell^0(\pi)} \underbrace{\sum_{\mathcal{D}_{\pi_0}} \overbrace{W(\pi_0 \rightarrow \pi; y)}^{\text{importance sampling}} \ell^0(s(x), \pi, y) d\pi}$$

$$= \underbrace{\int}_{\ell^0(\pi)} \underbrace{\sum_{\mathcal{D}_{1/2}} \overbrace{\ell^1(\pi, y)}^{\text{importance weight is } \ell^1}} \ell^0(s(x), \pi, y) d\pi$$

$$= \overbrace{\sum_{\mathcal{D}_{1/2}}}^{\text{swap}} \underbrace{\int}_{\ell^0(\pi)} \ell^1(\pi, y) \underbrace{\ell^0(s(x), \pi, y)} d\pi$$

$$= \sum_{\mathcal{D}_{1/2}} \overbrace{\int_{\ell^0(\pi) \cap \ell^0(s(x), \pi, y)}}^{\text{intersect}} \underbrace{\ell^1(\pi, y)} d\pi$$

$$= \sum_{\mathcal{D}_{1/2}} \overbrace{\ell^2(s(x), y)}^{\text{antidifferentiate}} d\pi$$

This flexible mathematical framework transforms a challenging integration problem into a tractable calculation using our core lemma. The key innovations (i.e., reframing importance weights and expressing classification correctness as set membership) provide deeper insight than the traditional integration-by-parts approach while generalizing beyond simple accuracy to various cost-sensitive losses. In the sections that follow, we demonstrate how this unified approach yields practical formulas for robust decision-making under class distribution uncertainty and asymmetric costs.

## D.1 Preliminaries

See Appendix B.2.1 for some of the notation used in this appendix.

**Definition D.1** (Adjusted Score).

$$s_{\pi'}(x) \triangleq \pi' \otimes s_{1/2}(x)$$

This represents the score optimally adjusted for the change in the prior probability. See Appendix B.2 for more details.

**Definition D.2** (The set of $1 - \pi$ for which $s$ gives the correct $\widehat{y}$).

$$\chi(x, y) \triangleq \begin{cases} [1 - y, s_{1/2}(x)] & \text{if } y = 1 \\ (s_{1/2}(x), 1 - y] & \text{if } y = 0 \end{cases}$$

For proof that these are the right conditions, see Lemma D.3.

**Lemma D.3** ($\chi$ is the right set).

$$1 - \pi \in \chi(x, y) \iff y = \kappa(\pi \otimes s_{1/2}(x), \tau)$$

*Proof.* We start by expressing the thresholding condition in terms of $1 - \pi$.

$$\pi \otimes (1 - \pi_0) \otimes s(x) \geq \tau$$

$$\implies (1 - \tau) \otimes (1 - \pi_0) \otimes s(x) \geq 1 - \pi$$

$$\implies s_{1/2}(x) \geq 1 - \pi$$

$$\implies 1 - \pi \in [0, s_{1/2}(x)]$$

Now we re-express the negation of this:

$$\pi \otimes s_{1/2}(x) \not\geq \tau$$

$$\implies 1 - \pi \notin [0, s_{1/2}(x)]$$

$$\implies 1 - \pi \in [s_{1/2}(x), 1)$$

We then compare to the label.

$$\begin{cases} \pi \otimes s_{1/2}(x) \geq \tau & \text{if } y = 1 \\ \pi \otimes s_{1/2}(x) < \tau & \text{if } y = 0 \end{cases}$$

$$\implies \begin{cases} 1 - \pi \in [1 - y, s_{1/2}(x)] & \text{if } y = 1 \\ 1 - \pi \in (s_{1/2}(x), 1 - y] & \text{if } y = 0 \end{cases}$$

$$\implies 1 - \pi \in \chi(x, y)$$

$\square$

**Definition D.4** (Clipping operator). Also called projection or restriction in other contexts.

$$\operatorname*{clip}_{[a,b]}(x) \triangleq \max(a, \min(b, x))$$

**Lemma D.5** (Intersecting $\chi$ with a closed interval). *This equality holds almost everywhere, because when the intersection is empty, the set on the right will be of measure zero. Integrals of bounded functions over these two sets will give the same results.*

$$[a, b] \cap \chi(x, y)$$

$$\overset{a.e.}{=} \begin{cases} [\operatorname*{clip}_{[a,b]}(1 - y), \ \operatorname*{clip}_{[a,b]}(s_{1/2}(x))] & \text{if } y = 1 \\ (\operatorname*{clip}_{[a,b]}(s_{1/2}(x)), \ \operatorname*{clip}_{[a,b]}(1 - y)] & \text{if } y = 0 \end{cases}$$

## D.2 Main Lemma

**Lemma D.6** (Average over Class Balance of a Sum over Data of a Cost-Weighted Correctness Condition).

$$(b - a) \underset{\pi \sim Uniform(a,b)}{\mathbb{E}} \underset{(x,y) \sim \mathcal{D}_\pi}{\mathbb{E}} C(\pi; y) \, \mathbf{1}_{\left(y = \kappa(\pi \otimes s_{1/2}(x), \tau)\right)}$$

$$= \underset{(x,y) \sim \mathcal{D}_{1/2}}{\mathbb{E}} \int_{p = \underset{[1-b,1-a]}{\mathrm{clip}}(s_{1/2}(x))}^{\underset{[1-b,1-a]}{\mathrm{clip}}(1-y)} 2(p - y)C(1 - p; y) \, dp$$

*Proof.*

$$\int_{\pi=a}^{b} \underset{(x,y) \sim \mathcal{D}_\pi}{\mathbb{E}} C(\pi; y) \, \mathbf{1}_{\left(y = \kappa(\pi \otimes s_{1/2}(x), \tau)\right)} d\pi$$

$$= \int_{\pi=a}^{b} \underset{(x,y) \sim \mathcal{D}_\pi}{\mathbb{E}} C(\pi; y) \, \mathbf{1}_{(1-\pi \in \chi(x,y))} d\pi$$

$$= \int_{\pi=a}^{b} \frac{1}{|\mathcal{D}_{\pi_0}|} \sum_{(x,y) \in \mathcal{D}_{\pi_0}} W(\pi_0 \to \pi; y) C(\pi; y) \, \mathbf{1}_{(1-\pi \in \chi(x,y))} d\pi$$

$$= \int_{\pi=a}^{b} \frac{1}{|\mathcal{D}_{\pi_0}|} \sum_{(x,y) \in \mathcal{D}_{\pi_0}} W(\pi_0 \to 1/2; y) \, 2 \, |(1 - \pi) - y| \, C(\pi; y) \, \mathbf{1}_{(1-\pi \in \chi(x,y))} d\pi$$

$$= \frac{1}{|\mathcal{D}_{\pi_0}|} \sum_{(x,y) \in \mathcal{D}_{\pi_0}} W(\pi_0 \to 1/2; y) \int_{\pi=a}^{b} 2|(1 - \pi) - y| C(\pi; y) \, \mathbf{1}_{(1-\pi \in \chi(x,y))} d\pi$$

We will now focus only on the inner integral.

$$\int_{\pi=a}^{b} 2|(1 - \pi) - y| C(\pi; y) \, \mathbf{1}_{(1-\pi \in \chi(x,y))} d\pi$$

$$= \int_{p=1-b}^{1-a} 2|p - y| C(1 - p; y) \, \mathbf{1}_{(p \in \chi(x,y))} dp$$

$$= \int_{[1-b,1-a] \cap \chi(x,y)} 2|p - y| C(1 - p; y) \, dp$$
by case analysis on y

$$= \int_{p = \underset{[1-b,1-a]}{\mathrm{clip}}(s_{1/2}(x))}^{\underset{[1-b,1-a]}{\mathrm{clip}}(1-y)} 2(p - y)C(1 - p; y) \, dp$$

Combining the two parts, we get the result.

$$(b-a) \underset{\pi \sim \text{Uniform}(a,b)}{\mathbb{E}} \underset{(x,y) \sim \mathcal{D}_{\pi}}{\mathbb{E}} C(\pi; y) \mathbf{1}_{\left(y = \kappa(\pi \otimes s_{1/2}(x), \tau)\right)}$$

$$= \int_{\pi=a}^{b} \frac{1}{|\mathcal{D}_{\pi_0}|} \sum_{(x,y) \in \mathcal{D}_{\pi}} C(\pi; y) \mathbf{1}_{\left(y = \kappa(\pi \otimes s_{1/2}(x), \tau)\right)} d\pi$$

$$= \frac{1}{|\mathcal{D}_{\pi_0}|} \sum_{(x,y) \in \mathcal{D}_{\pi_0}} W(\pi_0 \to 1/2; y) \int_{\pi=a}^{b} 2|(1-\pi) - y| C(\pi; y) \mathbf{1}_{(1-\pi \in \chi(x,y))} d\pi$$

$$= \frac{1}{|\mathcal{D}_{\pi_0}|} \sum_{(x,y) \in \mathcal{D}_{\pi_0}} W(\pi_0 \to 1/2; y) \int_{p=\underset{[1-b,1-a]}{\text{clip}}(s_{1/2}(x))}^{\underset{[1-b,1-a]}{\text{clip}}(1-y)} 2|p-y| C(1-p; y) \, dp$$

$$= \underset{(x,y) \sim \mathcal{D}_{1/2}}{\mathbb{E}} \int_{p=\underset{[1-b,1-a]}{\text{clip}}(s_{1/2}(x))}^{\underset{[1-b,1-a]}{\text{clip}}(1-y)} 2(p-y) C(1-p; y) \, dp$$

$\square$

## D.3 Example Use

**Theorem D.7** (Bounded Brier Score).

$$(b-a) \underset{\pi \sim [a,b]}{\mathbb{E}} PAMA(\mathcal{D}_{\pi}, s, \tau)$$

$$= \underset{(x,y) \sim \mathcal{D}_{1/2}}{\mathbb{E}} \left[ \left( \underset{[1-b,1-a]}{\text{clip}}(1-y) - y \right)^2 - \left( \underset{[1-b,1-a]}{\text{clip}}(s_{1/2}(x)) - y \right)^2 \right]$$

*Proof.* Straightforward application of Lemma D.6 with $C(\pi; y) = 1$.

$$(b - a) \underset{\pi \sim [a,b]}{\mathbb{E}} \text{PAMA}(\mathcal{D}_{\boldsymbol{\pi}}, s, \tau)$$

$$= (b - a) \underset{\pi \sim [a,b]}{\mathbb{E}} C(\pi; y) \, \mathbf{1}_{\left(y = \kappa(\pi \otimes s_{1/2}(x), \tau)\right)}$$

by Lemma D.6

$$= \underset{(x,y) \sim \mathcal{D}_{1/2}}{\mathbb{E}} \int_{p = \underset{[1-b,1-a]}{\text{clip}} (s_{1/2}(x))}^{\underset{[1-b,1-a]}{\text{clip}} (1-y)} 2(p - y) C(\pi; y) \, dp$$

$$= \underset{(x,y) \sim \mathcal{D}_{1/2}}{\mathbb{E}} \int_{p = \underset{[1-b,1-a]}{\text{clip}} (s_{1/2}(x))}^{\underset{[1-b,1-a]}{\text{clip}} (1-y)} 2(p - y) \, dp$$

by antidifferentiation

$$= \underset{(x,y) \sim \mathcal{D}_{1/2}}{\mathbb{E}} \left[ (p - y)^2 \right]_{p = \underset{[1-b,1-a]}{\text{clip}} (s_{1/2}(x))}^{p = \underset{[1-b,1-a]}{\text{clip}} (1-y)}$$

$$= \underset{(x,y) \sim \mathcal{D}_{1/2}}{\mathbb{E}} \left[ \left( \underset{[1-b,1-a]}{\text{clip}} (1 - y) - y \right)^2 - \left( \underset{[1-b,1-a]}{\text{clip}} (s_{1/2}(x)) - y \right)^2 \right]$$

$\square$

# E  Log Scores with Asymmetric Costs

**Theorem E.1** (Bounded Log Score)**.**

$$[\sigma^{-1}(b) - \sigma^{-1}(a)] \underset{\sigma^{-1}(\pi) \sim [\sigma^{-1}(a), \sigma^{-1}(b)]}{\mathbb{E}} PAMA(\mathcal{D}_{\boldsymbol{\pi}}, s, \tau)$$

$$= 2 \underset{(x,y) \sim \mathcal{D}_{1/2}}{\mathbb{E}} \log \left| \underset{[1-b,1-a]}{\text{clip}} (1 - y) - (1 - y) \right| - \log \left| \underset{[1-b,1-a]}{\text{clip}} (s_{1/2}(x)) - (1 - y) \right|$$

*Proof.* Use Lemma D.6 with $C(\pi; y) = \frac{1}{\pi(1-\pi)}$.

$$\left[\sigma^{-1}(b) - \sigma^{-1}(a)\right] \mathop{\mathbb{E}}_{\sigma^{-1}(\pi) \sim [\sigma^{-1}(a), \sigma^{-1}(b)]} \mathrm{PAMA}(\mathcal{D}_{\boldsymbol{\pi}}, s, \tau)$$

$$= [b - a] \mathop{\mathbb{E}}_{\pi \sim [a,b]} \mathop{\mathbb{E}}_{(x,y) \sim \mathcal{D}_{\boldsymbol{\pi}}} C(\pi; y) \, \mathbf{1}_{\left(y = \kappa(\pi \otimes s_{1/2}(x), \tau)\right)}$$

$$= \mathop{\mathbb{E}}_{(x,y) \sim \mathcal{D}_{\boldsymbol{1/2}}} \int_{p = \mathop{\mathrm{clip}}_{[1-b,1-a]}(s_{1/2}(x))}^{\mathop{\mathrm{clip}}_{[1-b,1-a]}(1-y)} 2(p - y) C(1 - p; y) \, dp \qquad \text{Lemma D.6}$$

$$= \mathop{\mathbb{E}}_{(x,y) \sim \mathcal{D}_{\boldsymbol{1/2}}} \int_{p = \mathop{\mathrm{clip}}_{[1-b,1-a]}(s_{1/2}(x))}^{\mathop{\mathrm{clip}}_{[1-b,1-a]}(1-y)} 2(p - y) \frac{dp}{(1 - p)p}$$

$$= \mathop{\mathbb{E}}_{(x,y) \sim \mathcal{D}_{\boldsymbol{1/2}}} \int_{p = \mathop{\mathrm{clip}}_{[1-b,1-a]}(s_{1/2}(x))}^{\mathop{\mathrm{clip}}_{[1-b,1-a]}(1-y)} \frac{2dp}{(1 - y - p)} \qquad \text{case analysis}$$

$$= 2 \mathop{\mathbb{E}}_{(x,y) \sim \mathcal{D}_{\boldsymbol{1/2}}} \ln \left| 1 - y - \mathop{\mathrm{clip}}_{[1-b,1-a]}(1 - y) \right| - \ln \left| 1 - y - \mathop{\mathrm{clip}}_{[1-b,1-a]} s_{1/2}(x) \right|$$

$\square$

## E.1 Weighted Accuracy

Refer to Definition C.7

**Theorem E.2** (PAMWA Log Score)**.**

$$\left[\sigma^{-1}(b) - \sigma^{-1}(a)\right] \mathop{\mathbb{E}}_{\sigma^{-1}(\pi) \sim [\sigma^{-1}(a), \sigma^{-1}(b)]} PAMWA(\mathcal{D}_{\boldsymbol{\pi}}, s, \tau, c)$$

$$= 2 \mathop{\mathbb{E}}_{(x,y) \sim \mathcal{D}_{\boldsymbol{1/2}}} \ln \left| 1 - y - \mathop{\mathrm{clip}}_{[c \otimes 1-b, c \otimes 1-a]}(1 - y) \right| - \ln \left| 1 - y - \mathop{\mathrm{clip}}_{[c \otimes 1-b, c \otimes 1-a]}(s_{1/2}(x)) \right|$$

*Proof.* The high level idea is we replace a PAMWA term with a PAMA term by Proposition C.8, and then do a change of variables, and apply Theorem E.1.

$$[\sigma^{-1}(b) - \sigma^{-1}(a)] \mathop{\mathbb{E}}_{\sigma^{-1}(\pi)\sim[\sigma^{-1}(a),\sigma^{-1}(b)]} \text{PAMWA}(\mathcal{D}_{\boldsymbol{\pi}}, s, \tau, c)$$

$$= [\sigma^{-1}(b) - \sigma^{-1}(a)] \mathop{\mathbb{E}}_{\sigma^{-1}(\pi)\sim[\sigma^{-1}(a),\sigma^{-1}(b)]} \text{PAMA}(\mathcal{D}_{1-c\otimes\pi}, s, \tau) \qquad \text{(by Proposition C.8)}$$

now we do a change of variables $\pi' = (1 - c) \otimes \pi$

$$= [\sigma^{-1}(b) - \sigma^{-1}(a)] \mathop{\mathbb{E}}_{\sigma^{-1}(\pi')\sim[\sigma^{-1}((1-c)\otimes a),\sigma^{-1}((1-c)\otimes b)]} \text{PAMA}(\mathcal{D}_{\pi'}, s, \tau)$$

$$= [\sigma^{-1}(1 - c \otimes b) - \sigma^{-1}(1 - c \otimes a)] \mathop{\mathbb{E}}_{\sigma^{-1}(\pi')\sim[\sigma^{-1}((1-c)\otimes a),\sigma^{-1}((1-c)\otimes b)]} \text{PAMA}(\mathcal{D}_{\pi'}, s, \tau)$$
$$\text{(by Proposition B.7)}$$

now by Theorem E.1

$$= 2 \mathop{\mathbb{E}}_{(x,y)\sim\mathcal{D}_{\mathbf{1/2}}} \ln \left| 1 - y - \mathop{\text{clip}}_{[1-(1-c)\otimes b, 1-(1-c)\otimes a]} (1 - y) \right| - \ln \left| 1 - y - \mathop{\text{clip}}_{[1-(1-c)\otimes b, 1-(1-c)\otimes a]} (s_{1/2}(x)) \right|$$

$$= 2 \mathop{\mathbb{E}}_{(x,y)\sim\mathcal{D}_{\mathbf{1/2}}} \ln \left| 1 - y - \mathop{\text{clip}}_{[c\otimes 1-b, c\otimes 1-a]} (1 - y) \right| - \ln \left| 1 - y - \mathop{\text{clip}}_{[c\otimes 1-b, c\otimes 1-a]} (s_{1/2}(x)) \right|$$

$\square$

## E.2  Net Benefit

Refer to Definition C.9

**Theorem E.3** (PAMNB Log Score)**.**

$$[\sigma^{-1}(b) - \sigma^{-1}(a)] \mathop{\mathbb{E}}_{\sigma^{-1}(\pi)\sim[\sigma^{-1}(a),\sigma^{-1}(b)]} PAMNB(\mathcal{D}_{\boldsymbol{\pi}}, s, \tau, c)$$

$$= \frac{2}{1 - c} \mathop{\mathbb{E}}_{(x,y)\sim\mathcal{D}_{(1-c)}} \ln \left| 1 - y - \mathop{\text{clip}}_{[1-b, 1-a]} (1 - y) \right| - \ln \left| 1 - y - \mathop{\text{clip}}_{[1-b, 1-a]} (s_{(1-c)}(x)) \right|$$

*Proof.*

$$\left(\sigma^{-1}(b) - \sigma^{-1}(a)\right) \underset{\sigma^{-1}(\pi) \sim [\sigma^{-1}(a), \sigma^{-1}(b)]}{\mathbb{E}} \text{PAMNB}(\boldsymbol{\mathcal{D}}_{\boldsymbol{\pi}}, s, \tau, c)$$

$$= \int_{\sigma^{-1}(\pi) = \sigma^{-1}(a)}^{\sigma^{-1}(b)} \text{PAMNB}(\boldsymbol{\mathcal{D}}_{\boldsymbol{\pi}}, s, \tau, c) \ d\sigma^{-1}(\pi)$$

$$= \int_{\pi=a}^{b} \text{PAMNB}(\boldsymbol{\mathcal{D}}_{\boldsymbol{\pi}}, s, \tau, c) \ \frac{d\pi}{\pi(1-\pi)}$$

$$= (b-a) \underset{\pi \sim [a,b]}{\mathbb{E}} \frac{\text{PAMNB}(\boldsymbol{\mathcal{D}}_{\boldsymbol{\pi}}, s, \tau, c)}{\pi(1-\pi)}$$

$$= (b-a) \underset{\pi \sim [a,b]}{\mathbb{E}} \underset{x,y \sim \boldsymbol{\mathcal{D}}_{\boldsymbol{\pi}}}{\mathbb{E}} \frac{\left(\frac{c}{1-c}\right)^{1-y} \mathbf{1}_{\left(y = \kappa(\pi \otimes s_{1/2}(x), c)\right)}}{\pi(1-\pi)}$$

$$= (b-a) \underset{\pi \sim [a,b]}{\mathbb{E}} \underset{x,y \sim \boldsymbol{\mathcal{D}}_{\boldsymbol{\pi}}}{\mathbb{E}} \frac{1}{\pi(1-\pi)} \frac{c^{1-y}(1-c)^y}{(1-c)} \mathbf{1}_{\left(y = \kappa(\pi \otimes (1-c) \otimes s_{1/2}(x), \tau)\right)}$$

now we use Lemma D.6

$$= \underset{(x,y) \sim \boldsymbol{\mathcal{D}}_{\mathbf{1/2}}}{\mathbb{E}} \int_{p = \underset{[1-b,1-a]}{\text{clip}}(1-c \otimes s_{1/2}(x))}^{\underset{[1-b,1-a]}{\text{clip}}(1-y)} 2(p-y) \frac{1}{(1-p)p} \frac{c^{1-y}(1-c)^y}{1-c} dp$$

$$= \frac{2}{1-c} \underset{(x,y) \sim \boldsymbol{\mathcal{D}}_{\mathbf{1/2}}}{\mathbb{E}} c^{1-y}(1-c)^y \int_{p = \underset{[1-b,1-a]}{\text{clip}}((1-c) \otimes s_{1/2}(x))}^{\underset{[1-b,1-a]}{\text{clip}}(1-y)} \frac{p-y}{(1-p)p} dp$$

and we use importance sampling

$$= \frac{2}{1-c} \underset{(x,y) \sim \boldsymbol{\mathcal{D}}_{\mathbf{1-c}}}{\mathbb{E}} \int_{p = \underset{[1-b,1-a]}{\text{clip}}(s_{1-c}(x))}^{\underset{[1-b,1-a]}{\text{clip}}(1-y)} \frac{p-y}{(1-p)p} dp$$

and by case analysis on y

$$= \frac{2}{1-c} \underset{(x,y) \sim \boldsymbol{\mathcal{D}}_{\mathbf{(1-c)}}}{\mathbb{E}} \int_{p = \underset{[1-b,1-a]}{\text{clip}}(s_{(1-c)}(x))}^{\underset{[1-b,1-a]}{\text{clip}}(1-y)} \frac{1}{1-y-p} dp$$

$$= \frac{2}{1-c} \underset{(x,y) \sim \boldsymbol{\mathcal{D}}_{\mathbf{(1-c)}}}{\mathbb{E}} \ln\left|1-y - \underset{[1-b,1-a]}{\text{clip}}(1-y)\right| - \ln\left|1-y - \underset{[1-b,1-a]}{\text{clip}}(s_{(1-c)}(x))\right|$$

$\square$

# F  DCA Brier Score

Although we believe the uniform log-odds for prevalence better fits clinical decision-making scenarios, we provide a derivation of the equivalent formulation using uniform prevalence for completeness.

**Theorem F.1** (PAMNB Brier Score)**.**

$$[b-a] \underset{\pi \sim [a,b]}{\mathbb{E}} \text{PAMNB}(\boldsymbol{\mathcal{D}}_{\boldsymbol{\pi}}, s, \tau, c)$$

$$= \frac{1}{2(1-c)} \underset{(x,y) \sim \boldsymbol{\mathcal{D}}_{\mathbf{1-c}}}{\mathbb{E}} \left[\left(\underset{[1-b,1-a]}{\text{clip}}(1-y) - y\right)^2 - \left(\underset{[1-b,1-a]}{\text{clip}}(s_{1-c}(x)) - y\right)^2\right]$$

*Proof.* Recall the definition of the adjusted score: $s_{\pi'}(x) \triangleq \pi' \otimes s_{1/2}(x)$.

And recall $s(x) > c \otimes \tau \iff (1-c) \otimes s(x) > \tau$

$$(b-a) \underset{\pi \sim [a,b]}{\mathbb{E}} \text{PAMNB}(\mathcal{D}_{\boldsymbol{\pi}}, s, \tau, c)$$

$$= (b-a) \underset{\pi \sim [a,b]}{\mathbb{E}} \underset{x,y \sim \mathcal{D}_{\boldsymbol{\pi}}}{\mathbb{E}} \left(\frac{c}{1-c}\right)^{1-y} \mathbf{1}_{\left(y=\kappa(\pi \otimes s_{1/2}(x),c)\right)}$$

$$= (b-a) \underset{\pi \sim [a,b]}{\mathbb{E}} \underset{x,y \sim \mathcal{D}_{\boldsymbol{\pi}}}{\mathbb{E}} \frac{c^{1-y}(1-c)^y}{(1-c)} \mathbf{1}_{\left(y=\kappa(\pi \otimes (1-c) \otimes s_{1/2}(x),\tau)\right)}$$

now we use Lemma D.6

$$= \underset{(x,y) \sim \mathcal{D}_{\boldsymbol{1/2}}}{\mathbb{E}} \int_{p=\underset{[1-b,1-a]}{\text{clip}}(1-c \otimes s_{1/2}(x))}^{\underset{[1-b,1-a]}{\text{clip}}(1-y)} 2(p-y) \frac{c^{1-y}(1-c)^y}{1-c} dp$$

$$= \frac{1}{1-c} \underset{(x,y) \sim \mathcal{D}_{\boldsymbol{1/2}}}{\mathbb{E}} c^{1-y}(1-c)^y \int_{p=\underset{[1-b,1-a]}{\text{clip}}((1-c) \otimes s_{1/2}(x))}^{\underset{[1-b,1-a]}{\text{clip}}(1-y)} 2(p-y) \ dp$$

and we use importance sampling

$$= \frac{1}{1-c} \underset{(x,y) \sim \mathcal{D}_{\boldsymbol{1-c}}}{\mathbb{E}} \frac{1}{2} \int_{p=\underset{[1-b,1-a]}{\text{clip}}(s_{1-c}(x))}^{\underset{[1-b,1-a]}{\text{clip}}(1-y)} 2(p-y) \ dp$$

by antidifferentiation

$$= \frac{1}{2(1-c)} \underset{(x,y) \sim \mathcal{D}_{\boldsymbol{1-c}}}{\mathbb{E}} \left[(p-y)^2\right]_{p=\underset{[1-b,1-a]}{\text{clip}}(s_{1-c}(x))}^{p=\underset{[1-b,1-a]}{\text{clip}}(1-y)}$$

$$= \frac{1}{2(1-c)} \underset{(x,y) \sim \mathcal{D}_{\boldsymbol{1-c}}}{\mathbb{E}} \left[\left(\underset{[1-b,1-a]}{\text{clip}}(1-y) - y\right)^2 - \left(\underset{[1-b,1-a]}{\text{clip}}(s_{1-c}(x)) - y\right)^2\right]$$

$\square$

# G   13 ways of looking at an AUC-ROC

For largely contingent historical reasons, medical informaticists have long reported results either in terms of specificity and sensitivity (if they want a fixed threshold), or in terms of the Area Under the Receiver Operating Characteristic Curve (AUC-ROC). What exactly does this mean? There are 12 standard interpretations of the AUC-ROC, and none of them fit very well. We add a thirteenth which is more relevant but still inadequate for our purposes.

1. 0.5 when the classifier is random, and 1.0 when the classifier is perfect. Carrington et al. [2023] reports this as the most common interpretation, a bit tongue in cheek. It is, unfortunately, also the current authors' experience that this is the most commonly given interpretation in practice.

2. The 2 alternative forced choice accuracy rate Swets and Birdsall [1956].
   This only makes sense in the original psychometric setting where an experimenter in fact guarantees that there is one positive and one negative case Carrington et al. [2023].

3. A rescaled version of the Mann-Whitney $U$ statistic Bamber [1975], Hanley and McNeil [1982]. This is actually the same as the statement above, but it sounds more impressive.

Note that AUC-ROC is never reported as a p-value based on this statistic, which suggests that the interpretation is not practically very useful.

4. A rescaled version of the Kendall's $\tau$ correlation coefficient Hernández-Orallo et al. [2013].

   It is technically true that the AUC-ROC is a pairwise permutation distance between the ideal ranking and the actual ranking. But there are only 2 ranks! This makes the exercise meaningless.

5. An average of precision (though not "Average Precision" which refers to something else)

   There is an occasional attempt to rescue the paradigm by arguing that AUC-ROC shows an average of $TP = K \times$ Precision@$K$ over a range of $K$. The trouble is twofold:

   - This gives a uniform average over all possible values of $K$. In a quantity constrained setting where we're forced to pick out only $K$ items to give positive labels (imagine only 10 doses of penicillin in the freezer) we generally have very small K.
   - Few binary classification problems are actually intended for explicit quantity constraints, so this doesn't fit well.

6. An average of power over a range of sizes (in the Neyman-Pearson sense)

   This one is very popular with practitioners and virtually absent from the literature, aside from McClish [1989], which was later criticized by McClish [2012], Carrington et al. [2023]. The trouble is that in the Neyman-Pearson paradigm we're meant to pick a power, and then find out what the size of the test is. Even if we reverse this interpretation as an average size over a range of powers, the power in the NP paradigm isn't an empirical quantity we might observe a distribution over. We're meant to pick one.

7. It is the area under a curve if FPR is plotted against TPR. Hanley and McNeil [1982]

8. Average accuracy on the positive class across a uniform distribution of accuracy on the negative class, or vice versa Metz [1986, 1989], Zhou et al. [2002]. This is actually the same as the statement above, but more useful-sounding, and slightly less mysterious since it doesn't use the words "False Positive Rate" or "True Positive Rate".

9. Given two thresholds $a < b$, the average accuracy on the positive class across a uniform distribution of accuracy on the negative class between those two thresholds, plus the average accuracy on the negative class across a uniform distribution of accuracy on the positive class between those two thresholds, weighted by the class balance between those two thresholds Carrington et al. [2023]. This is a bit more useful, but it's not clear why we're using the exact class balance between the thresholds.

10. An average of accuracy as we set the threshold at each data point, leaving operating conditions the same Flach et al. [2011]. This interpretation is the only one from this set of authors that directly addresses the problem that data is empirical and discrete.

11. An average of cost-weighted error over a range of cost ratios. Hand [2009] shows that if a score is calibrated, then the AUC-ROC is an average of the cost-weighted error over a range of prevalences. The trouble is that:

    - Calibration is a really important property of a score! Without it, we can only make top-K decisions. Even then we have to assume there are more than $K$ real positives.
    - The range of costs is a function of model scores on the training data. But the risk that a given individual has syphilis is not a good estimator of how much less harmful unnecessary penicillin is than untreated tertiary syphilis.
    - Because the costs are a function of the model, if two people train models on the same data, they will get different costs, and the average accuracies will be incommensurable.

12. An average of skew-weighted cost, for skew $z = c \otimes (1 - \pi)$. Hernández-Orallo et al. [2013] proposes this interpretation, although again the costs are set arbitrarily and distinctly by each model, and class-conditional distributions are assumed to be continuous and fully known. Furthermore, it is not clear what it means to integrate over values of $c \otimes (1 - \pi)$. One possible interpretation is that we set $c = (1 - \pi)$ and then calculate a line integral along a particular curve in cost / prevalence space, by way of estimating the area integral over the entire space.

13. An average of accuracy under label shift, where the distribution of positive class prevalences is derived by sampling from the model scores on the training data. This is syntactically

similar, but specifically derived in the case of the sampling problems that arise from label shift (see Theorem H.5). While this is the most relevant definition for our problem, it shares the issues of the distribution being unrelated to the class balance of the deployment and incommensurable across models.

# H  AUC-ROC as Average Accuracy under Label Shift (symmetry, no precision)

See Appendix G for an enumeration of alternative interpretations of the AUC-ROC, an idea inspired by Turakhia [2017], Stevens [1923]. Here we focus on the interpretation as an average of shifted accuracy, when the class balance distribution is derived by taking one minus the score from samples drawn from the training data, assuming that the classifier is calibrated, so that $\mathbb{P}(y = 1 \mid s(x) = \tau) = \tau$ for all $\tau \in [0, 1]$.

As discussed in Section 3, this interpretation shows a clear meaning for AUC-ROC in our context, but also shows its shortcomings.

Our key innovation is to focus on using importance sampling to directly address label shift. We begin with a definition, which we then reexpress in terms of balanced classes. Then we use importance sampling to reexpress components of the AUC-ROC in terms of true positives or true negatives. Finally, thanks to the intrinsic class symmetry of the AUC-ROC, we combine the results.

**Definition H.1** (AUC-ROC as Average over Data)**.**

$$\text{AUC-ROC}(\mathcal{D}_{\pi_0}, s) \triangleq \sum_{(x,y) \in \mathcal{D}_{\pi_0}} \frac{1}{|\mathcal{D}_{\pi_0}|} \frac{1-y}{1-\pi_0} \sum_{(x',y') \in \mathcal{D}_{\pi_0}} \frac{1}{|\mathcal{D}_{\pi_0}|} \frac{y'}{\pi_0} \Big[ \mathbf{1}_{(s(x')>s(x))} + {}^{1\!}/{}_2 \mathbf{1}_{(s(x')=s(x))} \Big]$$

We now show that this is equivalent to drawing from a class-balanced distribution, a known property of the AUC-ROC, but one which will simplify our analysis.

**Lemma H.2** (Importance Weighting)**.**

$$\frac{1}{|\mathcal{D}_{\pi_0}|^2} \sum_{(x,y) \in \mathcal{D}_{\pi_0}} \sum_{(x',y') \in \mathcal{D}_{\pi_0}} \frac{1-y}{1-\pi_0} \frac{y'}{\pi_0} \Big[ \mathbf{1}_{(s(x')>s(x))} + {}^{1\!}/{}_2 \mathbf{1}_{(s(x')=s(x))} \Big]$$

$$= \mathop{\mathbb{E}}_{(x,y) \in \mathcal{D}_{1/2}} \mathop{\mathbb{E}}_{(x',y') \in \mathcal{D}_{1/2}} (1-y)y' \Big[ \mathbf{1}_{(s(x')>s(x))} + {}^{1\!}/{}_2 \mathbf{1}_{(s(x')=s(x))} \Big]$$

In the continuous case, the ${}^{1\!}/{}_2 \mathbf{1}_{(s(x')=s(x))}$ term disappears. In the discrete case, the AUC-ROC is neither exactly equal to the average of true positives nor the average of true negatives, but instead to the average of the two.

**Lemma H.3** (Average over Label Shifts of True Positive Rate)**.**

$$\mathop{\mathbb{E}}_{(x,y) \in \mathcal{D}_{1/2}} \mathop{\mathbb{E}}_{(x',y') \in \mathcal{D}_{1/2}} (1-y)y' \Big[ \mathbf{1}_{(s(x')>s(x))} + \mathbf{1}_{(s(x')=s(x))} \Big]$$

$$\mathop{\mathbb{E}}_{t \in s[\mathcal{D}_{1/2}]} \mathop{\mathbb{E}}_{(x',y') \in \mathcal{D}_{1/2}} \Big[ \mathop{\mathbb{E}}_{(x,y) \in \mathcal{D}_{1/2}:s(x)=t} 1-y \Big] y' \Big[ \mathbf{1}_{(s(x') \geq t)} \Big]$$

$$= \mathop{\mathbb{E}}_{t \in s[\mathcal{D}_{1/2}]} \Big[ \mathop{\mathbb{E}}_{(x,y) \in \mathcal{D}_{1/2}} \mathbf{1}_{(s(x)=t)} \Big] \mathop{\mathbb{E}}_{(x',y') \in \mathcal{D}_{1/2}} (1-t) \cdot y' \Big[ \mathbf{1}_{(s(x') \geq t)} \Big]$$

$$= \mathop{\mathbb{E}}_{t \in s[\mathcal{D}_{1/2}]} \mathop{\mathbb{E}}_{(x',y') \in \mathcal{D}_{1-t}} y' \cdot \mathbf{1}_{(s(x') \geq t)}$$

By reversing the order of summation, we can also express a related quantity as an average of true negatives, instead.

**Lemma H.4** (Average over Label Shifts of True Negative Rate).

$$\mathop{\mathbb{E}}_{(x,y)\in\mathcal{D}_{1/2}}\mathop{\mathbb{E}}_{(x',y')\in\mathcal{D}_{1/2}}(1-y)y'\Big[\mathbf{1}_{(s(x')>s(x))}\Big]$$

$$=\mathop{\mathbb{E}}_{(x',y')\in\mathcal{D}_{1/2}}\mathop{\mathbb{E}}_{(x,y)\in\mathcal{D}_{1/2}}y'(1-y)\Big[\mathbf{1}_{(s(x')>s(x))}\Big]$$

$$=\mathop{\mathbb{E}}_{t\in s[\mathcal{D}_{1/2}]}\mathop{\mathbb{E}}_{(x,y)\in\mathcal{D}_{1/2}}\left[\mathop{\mathbb{E}}_{(x',y')\in\mathcal{D}_{1/2}:s(x')=t}y'\right](1-y)\Big[\mathbf{1}_{(t>s(x))}\Big]$$

$$=\mathop{\mathbb{E}}_{t\in s[\mathcal{D}_{1/2}]}\left[\mathop{\mathbb{E}}_{(x',y')\in\mathcal{D}_{1/2}}\mathbf{1}_{(s(x')=t)}\right]\mathop{\mathbb{E}}_{(x',y')\in\mathcal{D}_{1/2}}t(1-y)\Big[\mathbf{1}_{(t>s(x))}\Big]$$

$$=\mathop{\mathbb{E}}_{t\in s[\mathcal{D}_{1/2}]}\mathop{\mathbb{E}}_{(x,y)\in\mathcal{D}_{1-t}}(1-y)\cdot\mathbf{1}_{(s(x)<t)}$$

Finally, we combine these three results to show that the AUC-ROC is an average of accuracy under label shift.

**Theorem H.5** (AUC-ROC As Average Accuracy).

$$2AUC\text{-}ROC(s)=\mathop{\mathbb{E}}_{t\sim s[\mathcal{D}_{1/2}]}PAMA(\mathcal{D}_{1-t},s,1/2)$$

*Proof.*

$$2\text{AUC-ROC}(\mathcal{D}_{\pi_0},s)$$

$$=\mathop{\mathbb{E}}_{(x,y)\in\mathcal{D}_{1/2}}\mathop{\mathbb{E}}_{(x',y')\in\mathcal{D}_{1/2}}(1-y)y'\Big[2\cdot\mathbf{1}_{(s(x')>s(x))}+\mathbf{1}_{(s(x')=s(x))}\Big]$$

$$=\mathop{\mathbb{E}}_{(x,y)\in\mathcal{D}_{1/2}}\mathop{\mathbb{E}}_{(x',y')\in\mathcal{D}_{1/2}}(1-y)y'\Big[\mathbf{1}_{(s(x')>s(x))}+\mathbf{1}_{(s(x')=s(x))}\Big]$$

$$+\mathop{\mathbb{E}}_{(x,y)\in\mathcal{D}_{1/2}}\mathop{\mathbb{E}}_{(x',y')\in\mathcal{D}_{1/2}}(1-y)y'\Big[\mathbf{1}_{(s(x')>s(x))}\Big]$$

$$=\mathop{\mathbb{E}}_{t\sim s[\mathcal{D}_{1/2}]}\left[\mathop{\mathbb{E}}_{(x',y')\in\mathcal{D}_{1-t}}y'\cdot\mathbf{1}_{(s(x')\geq t)}+\mathop{\mathbb{E}}_{(x,y)\in\mathcal{D}_{1-t}}(1-y)\cdot\mathbf{1}_{(s(x)<t)}\right]$$

$$=\mathop{\mathbb{E}}_{t\sim s[\mathcal{D}_{1/2}]}[\text{Accuracy}(\mathcal{D}_{1-t},s(x),\tau=t)]$$

$$=\mathop{\mathbb{E}}_{t\sim s[\mathcal{D}_{1/2}]}[\text{Accuracy}(\mathcal{D}_{1-t},1-t\otimes s(x),\tau=1/2)]$$

$$=\mathop{\mathbb{E}}_{t\sim s[\mathcal{D}_{1/2}]}[\text{PAMA}(\mathcal{D}_{1-t},s(x),\tau=1/2)]$$

$\square$

# I  eICU Subgroup Analysis

## I.1  Patient Demographics

Table 12: Baseline Characteristics and Outcomes by Sex and Race/Ethnicity

| Characteristic | Overall | Female | Male | African-American | Caucasian |
|---|---|---|---|---|---|
| N | 2520 | 1008 | 1508 | 231 | 2010 |
| Age (years), Mean (SD) | 63.3 (17.8) | 64.7 (18.2) | 62.3 (17.4) | 56.0 (16.8) | 64.5 (17.5) |
| APACHE IV, Mean (SD) | 52.0 (25.3) | 54.3 (25.3) | 50.5 (25.1) | 50.4 (27.7) | 52.0 (24.8) |
| ICU LOS (days), Median | 1.5 | 1.5 | 1.5 | 1.6 | 1.4 |
| Hospital LOS (days), Median | 4.4 | 4.8 | 4.3 | 5.2 | 4.3 |
| Hospital Mortality, N (%) | 212 (8.4%) | 87 (8.6%) | 124 (8.2%) | 9 (3.9%) | 173 (8.6%) |
| ICU Mortality, N (%) | 126 (5.0%) | 42 (4.2%) | 83 (5.5%) | 7 (3.0%) | 101 (5.0%) |

## I.2 Mechanism Shift vs Label Shift

To separate which parts of the difference in log score are due to different class balance vs different mechanism of prediction, we can rely on the label shift assumption (within subgroups) and use the do calculus of Pearl [2001]

$$
\begin{array}{cc}
\mathcal{D} & \mathcal{D} \\
\downarrow \searrow & \searrow \\
Y \to X \to \kappa & \text{do } Y \to X \to \kappa \\
(1) & (2)
\end{array}
$$

Figure 3: Causal diagrams: (1) shows differences in performance are based both on label shift and differences in mechanism between subgroups (2) if we can intervene on Y to hold it constant, then we can measure differences in performance based purely on differing performance of the model between subgroups. As covered in the main body of the text, under label shift, we can simulate changes in Y using importance weighting and prior-adjusted maximal accuracy.

For brevity, all our expectations over $\pi$ in this section will assume the distribution

$$
\sigma^{-1}(\pi) \sim \text{Uniform}(\sigma^{-1}(\pi_{\text{blue}}), \sigma^{-1}(\pi_{\text{orange}}))
$$

The causal impact after intervention on Y is given by the clipped cross entropy:

$$
\Delta_{\mathcal{D} \to X} = \mathbb{E}_{\pi} \text{PAMNB}(\mathcal{D}_{\text{orange},\pi}, s, c) - \text{PAMNB}(\mathcal{D}_{\text{blue},\pi}, s, c)
$$

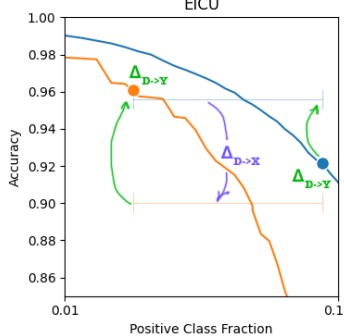

We can now additively decompose the difference in Accuracy:

$$
\Delta_{\mathcal{D} \to \kappa} = \Delta_{\mathcal{D} \to X} + \Delta_{\mathcal{D} \to Y}
$$

Which allows us to calculate the label shift effect:

$$
\Delta_{\mathcal{D} \to Y} = \mathbb{E}_{\pi} \left[ \left[ \text{PAMNB}(\mathcal{D}_{\text{orange},\pi}, s, c) - \text{PAMNB}(\mathcal{D}_{\text{orange}}, s, c) \right] \right.
$$

$$
\left. + \left[ \text{PAMNB}(\mathcal{D}_{\text{blue}}, s, c) - \text{PAMNB}(\mathcal{D}_{\text{blue},\pi}, s, c) \right] \right]
$$

We can think of these two terms as the difference between actual accuracy and average accuracy for the two separate subgroups.

### I.3 Calibration vs Sharpness

We can also decompose the difference in accuracy into calibration loss and sharpness.

$$\Delta_{TOTAL} = \Delta_S + \Delta_C$$

It's easy to calculate the overall gap.

$$\Delta_{TOTAL} = \mathbb{E}_{\pi} \, \text{PAMNB}(\mathcal{D}_{\text{orange},\pi}, s, c) - \text{PAMNB}(\mathcal{D}_{\text{blue},\pi}, s, c)$$

The sharpness gap is also straightforward, since we can simply recalibrate the model on the subgroup of the evaluation set (which we'll call $s^*$), and measure the loss of the recalibrated model.

$$\Delta_S = \mathbb{E}_{\pi} \, \text{PAMNB}(\mathcal{D}_{\text{orange},\pi}, s^*_{\text{orange}}, c) - \text{PAMNB}(\mathcal{D}_{\text{blue},\pi}, s^*_{\text{blue}}, c)$$

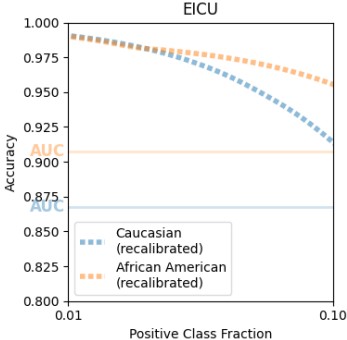

Figure 4: Accuracy vs Prevalence curves for subgroup-recalibrated models (dashed lines) plotted on the same y-axis as AUC (solid lines)

The difference in sharpness for clipped cross-entropy is not exactly the same as the difference in sharpness measured by the AUC-ROC because, as mentioned earlier, the AUC-ROC weights different thresholds differently. However, the two are fairly correlated. Sharpness across the full range of prevalences is somewhat higher for African Americans (orange).

This remaining gap is the calibration gap.

$$\Delta_C = \mathbb{E}_{\pi} \left[ \left[ \text{PAMNB}(\mathcal{D}_{\text{orange},\pi}, s, c) - \text{PAMNB}(\mathcal{D}_{\text{orange},\pi}, s^*_{\text{orange}}, c) \right] \right.$$
$$\left. + \left[ \text{PAMNB}(\mathcal{D}_{\text{blue},\pi}, s^*_{\text{blue}}, c) - \text{PAMNB}(\mathcal{D}_{\text{blue},\pi}, s, c) \right] \right]$$

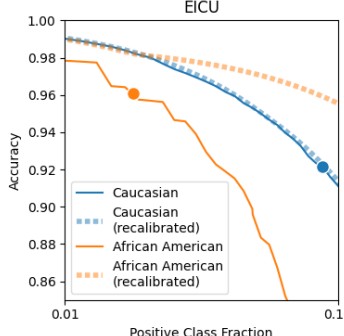

Figure 5: Accuracy vs Prevalence curves for original and recalibrated models

We can measure the miscalibration of the model by the difference in accuracy between the original and recalibrated versions. We plot the accuracy of the original model (solid lines) and the accuracy of a perfectly recalibrated model (dashed lines). Miscalibration for Caucasian patients (gap between blue lines) is essentially zero, while miscalibration for African American patients (gap between orange lines) is quite large.

The solid lines combine both the sharpness and miscalibration losses onto a single axis. There is no equivalent way to plot ECE on this y-axis.

### I.4 The Importance of Bounding the Prevalance Range

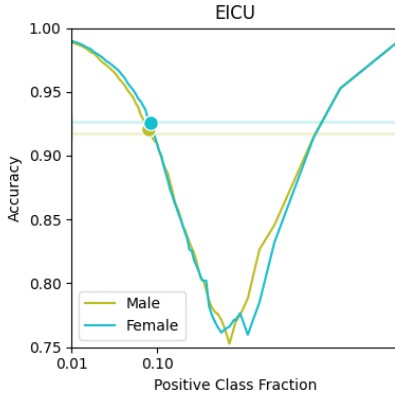

Accuracy vs. prevalence curves by gender. Accuracy at clinically relevant (low) prevalences is higher for female patients (blue) than for male patients (green).

However, at roughly balanced or high prevalences—ranges that are not clinically meaningful—the trend reverses slightly, with higher accuracy for males. Traditional scoring rules such as cross-entropy overweight these high-prevalence regions, reporting a better overall score for male patients despite inferior clinical performance at realistic prevalences. By contrast, the Bounded DCA Log Score correctly emphasizes clinically plausible ranges and thus ranks the model higher for female patients.

Figure 6: Accuracy vs Prevalence curves by gender

### I.5 Confidence Intervals

As is often the case, it's easy to take this too seriously. Since the prevalence of mortality is low, and the fraction of black patients is already low as well, it turns out there's a tremendous amount of sampling error in the public subset. When we draw the confidence intervals, we see that we can't really conclude anything.

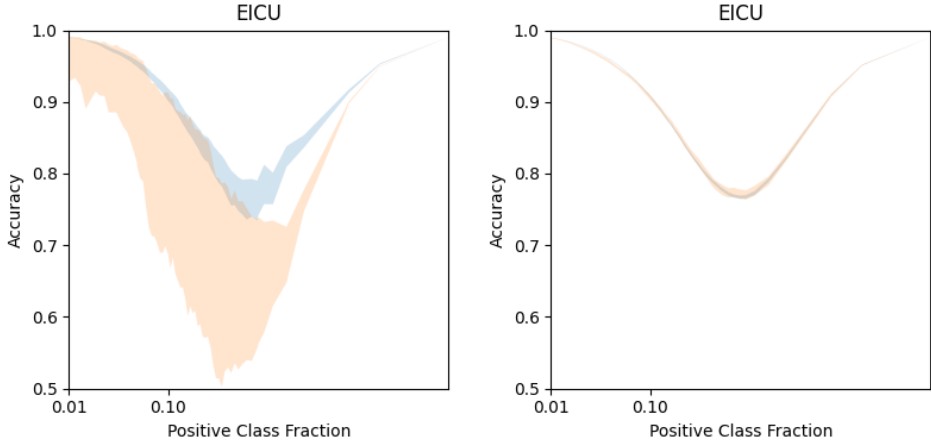

Figure 7: There's a significant gap between the performance on the public dataset for black and white patients, but it's well within the 95% confidence interval. By comparison, when we use the full dataset, we see that the quality of the models is actually quite similar, and any difference in clinical utility is largely due to the difference in prevalence.

### I.6 Conclusion

While AUC-ROC and Accuracy each handle some aspects of performance, we show an example with well known public data where both give the same misleading intuition about subgroup performance differences. What makes the clipped cross entropy approach useful in this setting is that its linearity makes it easy to decompose different effects, and its integral across scenarios allows us to catch effects that can't be seen at any one class balance.

It is also important that in each of these decomposition plots the two effects we were trying to decompose were measured in the same units, so we don't have to ask ourselves which one is more important. This generalizes to Net Benefit and Weighted Accuracy decompositions as well.

## J   Metric Failure Mode Comparison

| Metric | Domain Adaptation | Failure Mode Cost-Sensitive Evaluation | Decomposability | Notes |
|---|---|---|---|---|
| **Accuracy** | Does not handle subgroup prevalence shifts. | Assumes symmetric costs (FP = FN) | Does not reflect calibration. Very little information on discrimination. | Simpson's paradox effects when subgroup prevalences differ. |
| **AUC-ROC** | Invariant to prevalence. | Range of costs chosen by model, not clinicians. | Pure discrimination, no calibration. | Can't combine with (e.g.) ECE. |
| **Expected Calibration Error (ECE)** | Domain-specific adaptation unclear. | No cost sensitivity. | Pure calibration, no discrimination. | Can't combine with (e.g.) AUC-ROC. |
| **Brier Score** | Unrealistic prevalence ranges (e.g., half of weight above 50% mortality). | Cannot combine asymmetric costs with imbalanced classes. | Decomposes into calibration and discrimination. | Misleading at clinical prevalences. |
| **Cross-Entropy / NLL** | Unrealistic prevalence ranges (e.g., half of weight above 50% mortality). | Cannot combine asymmetric costs with imbalanced classes. | Decomposes into calibration and discrimination. | Misleading at clinically relevant prevalences. |
| **Net Benefit** | Requires known prevalence. | Requires known costs. | No decomposition into calibration and discrimination. | Requires exact knowledge. |

Table 13: Summary of metric failures under different evaluation scenarios.

