# OpenReview forum: "Aligning Evaluation with Clinical Priorities: Calibration, Label Shift, and Error Costs"
_NeurIPS.cc/2025/Conference — NeurIPS 2025 poster_

### Official Review · Reviewer_syGE · 2025-06-19

**Clarity:** 4
**Significance:** 3
**Originality:** 3
**Rating:** 6
**Confidence:** 3

**Summary:**

Widely used performance scores like accuracy and AUC-ROC often fail to adequately capture key clinical priorities. The authors propose an evaluation framework that explicitly incorporates calibrated thresholded classifiers, leveraging uncertainty for in-class prevalences and domain-specific cost asymmetries. They adapt a model originally developed for weather prediction, based on Schervish representation. They extend this framework to accommodate label shift and asymmetric costs, and to average cost-sensitive metrics over a bounded range of class balances.

**Questions:**

### Section 5 - application to subgroup decomposition
- This section lacks a brief description of the eICU dataset beyond the task of predicting in-hospital mortality using APACHE IV scores. I recommend including contextual information about the clinical setting, the types of clinical variables available, what the APACHE IV scores represent, the number of patients, patient demographics, the class distribution, etc.
- It is unclear how the evaluation metrics were obtained, as the authors do not describe the model training process. Key experimental details are missing.
- Typically, AUC-ROC curves plot the true positive rate (TPR) on the y-axis and the false positive rate (FPR) on the x-axis. However, the authors use “Accuracy” on the y-axis and “Positive Class Fraction” on the x-axis in their figures. This choice should be clarified. Additionally, the authors state that “(C) African American patients (orange) have noticeably better AUC-ROC than Caucasian patients (blue),” but the figures appear to show the opposite.
- The analysis of the eICU dataset is very brief and would benefit from further elaboration.

**Discussion**: It is unclear how the proposed method would perform in real-world clinical settings, especially when the class distribution at deployment is unknown. Based on lines 369–371, it appears that prior knowledge of prevalence is required before deployment, which is typically not available in practice.

**Code Availability**: The authors provide the code in the Supplementary Material, but it would be more practical and reproducible to host it in an anonymized GitHub repository. Additionally, the code lacks documentation, such as a README file, which makes it harder to understand and use. However, the code itself is clean and well structured.

Minor comments:
- Punctuation at the end of paragraph header is missing in some cases, and consistently in the Appendix.
- Figures in page 8 are missing numbers, and they should be referenced in the manuscript.

**Ethical Concerns:**

["NO or VERY MINOR ethics concerns only"]

**Final Justification:**

I am satisfied with the authors’ responses regarding binary classification, prevalence, and their extended evaluation.

Most of my concerns have been addressed, and after considering the other reviewers' comments along with the authors’ rebuttal, I have decided to increase my score.

**Limitations:**

Yes

**Quality:**

4

**Strengths And Weaknesses:**

Strengths:
- The paper is well written and structured. However, I find confusing that there is a "Related work" section, but background/related work spans both Sections 3 and 4.
- The methodology is well formulated, with justifications appropriately referenced and detailed in the appendix.
- The discussion section is well organized, clearly presenting limitations and outlining directions for future work.

Weaknesses:
- Section 5 lacks important experimental details, including dataset descriptions and model training procedures. The discussion of the results is very brief.
- The proposed method appears to be designed for binary classification. However, in medical contexts, this may be too limiting. Risk stratification often requires multiple categories (e.g., low, medium, high) to prioritize treatment. Is the framework restricted to binary classification, or could it be extended to multi-class settings?
- The method assumes that prevalence is known prior to deployment, which is often not the case in real-world applications.

---

> ### Author Rebuttal · Authors · 2025-07-29
>
> # Introduction: New Analysis and Major Commitments for Revision
>
> We thank the reviewers for thoughtfully highlighting the value of more explicit comparisons of our proposed metric to popular alternatives on real data.  To that end, we have extended the analysis from Section 5 on the performance of the Apache IV mortality prediction algorithm in different subgroups of the publicly available eICU dataset, and will include the following tables in the revised draft.
>
> | Metric | Caucasian | African-American |
> |--------|-----------|------------------|
> | Bounded DCA Log Score (calibration-only) | **0.999** | 0.927 |
> | ECE | **0.829** | 0.889 |
> | Bounded DCA Log Score (discrimination-only) | 0.957 | **0.973** |
> | AUC-ROC | 0.868 | **0.907** |
> | Bounded DCA Log Score | **0.956** | 0.900 |
> | ECE & AUC-ROC | ? | ? |
>
> *Table 1: Calibration and Discrimination for Caucasian and African-American Patients*
>
> Both commonly used and proposed calibration measures agree that the model has better calibration for African-American patients than Caucasian, but worse discrimination.
> However, the calibration and discrimination components for Bounded DCA Log Score can be linearly combined, while there is no principled way to combine AUC-ROC with ECE to reach an overall judgment.
>
> | Metric | Caucasian | African-American |
> |--------|-----------|------------------|
> | Accuracy Curve@1% Prevalence | **0.990** | 0.978 |
> | Accuracy Curve@5% Prevalence | **0.953** | 0.896 |
> | Accuracy Curve@10% Prevalence | **0.912** | 0.793 |
> | Bounded DCA Log Score | **0.956** | 0.900 |
> | Accuracy | 0.922 | **0.961** |
>
> *Table 2: Accuracy for Caucasian and African-American patients at varying prevalences*
>
> The most powerful analysis technique here is to graph accuracy at each prevalence and compare the two entire curves visually. As the table suggests, Apache IV on the public eICU data performs better for Caucasian patients than for African-American patients at any given prevalence.
> Comparing two Bounded DCA Log Scores averages the heights of these curves and the scalar summary preserves their ranking.
> The empirical accuracy for each subgroup is measured at different prevalences.
>
> | Metric | Male | Female |
> |------------|------|--------|
> | Accuracy@1% | 0.989 | 0.990 |
> | Accuracy@5% | 0.948 | 0.954 |
> | Accuracy@25% | 0.818 | 0.814 |
> | Accuracy@75% | 0.846 | 0.832 |
> | Accuracy@95% | 0.953 | 0.953 |
> | Accuracy@99% | 0.990 | 0.990 |
>
> *Table 3: Accuracy for Male and Female Patients*
>
> Apache IV performs better for female patients than male patients at low mortality prevalences, worse for roughly balanced mortality, and about the same at high mortality prevalences. However, only the low prevalence ranges are clinically relevant.
>
> | Metric | Male | Female |
> |--------|------|--------|
> | Brier | 0.062 | 0.062 |
> | Cross-Entropy / NLL | 0.214 | 0.218 |
> | Bounded DCA Log Score | 0.917 | 0.926 |
>
> *Table 4: Scoring Rules for Male and Female Patients*
>
> Although unbounded Brier score and cross-entropy are related to integrals of net cost, and are therefore related to net benefit, they give misleading answers because they put too much weight at high, clinically unrealistic prevalences.
>
>
>
> # Specific Reviewer Questions and Replies
>
> ## Question 0. The proposed method appears to be designed for binary classification. However, in medical contexts, this may be too limiting. Risk stratification often requires multiple categories (e.g., low, medium, high) to prioritize treatment. Is the framework restricted to binary classification, or could it be extended to multi-class settings?
>
> The framework we present is only valid in binary classification settings.
> The current binary scoring rule literature relies on properties of extremal convex functions which do not generalize to higher dimensions [4].
> If labels are discrete with total ordering (e.g. low, medium, high), Brier and log scores can be extended using ranked probability scores [1].
> If labels form a metric space, ideas like Hyvarinen's rule [2] and continuous ranked probability score [3] apply.
> We are studying other approaches.
>
> ## Question 1. This section lacks a brief description of the eICU dataset beyond the task of predicting in-hospital mortality using APACHE IV scores.
>
> > I recommend including contextual information about the clinical setting, the types of clinical variables available, what the APACHE IV scores represent, the number of patients, patient demographics, the class distribution, etc.
>
> We will include more detail on APACHE IV [5], the eICU data [6] [7], and descriptive statistics for both datasets in the next draft, as mentioned in our new analysis section.
>
> ## Question 2. It is unclear how the evaluation metrics were obtained, as the authors do not describe the model training process. Key experimental details are missing.
>
> To make reproducibility simple, we did not train new models.
> Instead, we used the Apache IV mortality predictions recorded in the eICU dataset itself; these are available for direct download from the PhysioNet website (links are not allowed in rebuttals this year).
>
> ## Question 3. Figure label issues
>
> > Typically, AUC-ROC curves plot the true positive rate (TPR) on the y-axis and the false positive rate (FPR) on the x-axis.
> However, the authors use "Accuracy" on the y-axis and "Positive Class Fraction" on the x-axis in their figures.
> This choice should be clarified.
> Additionally, the authors state that "(C) African American patients (orange) have noticeably better AUC-ROC than Caucasian patients (blue)," but the figures appear to show the opposite.
>
> As we will indicate more clearly in the revision, these are not ROC plots.
> We will also more explicitly explain what is being shown.
>
> ## Question 4. The analysis of the eICU dataset is very brief and would benefit from further elaboration.
>
> We will expand the analysis, including the four tables mentioned in the introduction and providing clearer exposition of the figures.
>
> ## Question 5 (Discussion).
>
> > It is unclear how the proposed method would perform in real-world clinical settings, especially when the class distribution at deployment is unknown.
> Based on lines 369–371, it appears that prior knowledge of prevalence is required before deployment, which is typically not available in practice.
>
> Bounded DCA log score is intended for use when prevalence can be bounded *but not precisely specified.*
> In the limit where bounds coincide, it becomes net benefit.
> However, the dearth of exact prevalence knowledge appears to be why net benefit-based approaches have not been more widely adopted.
> We show that non-clipped log score and Brier score are equivalent to integrating over the full interval between 0 and 1.
> However, in clinical settings, loose bounds are often available.
> For example, in this dataset, we can bound ICU patients' mortality between one in 100 and one in 10. As we show in Tables 3 and 4, those bounds are sufficient to improve classifier evaluation clinical utility.
>
> ## Question 6 (Code Availability).
>
> > The authors provide the code in the Supplementary Material, but it would be more practical and reproducible to host it in an anonymized GitHub repository. Additionally, the code lacks documentation, such as a README file, which makes it harder to understand and use. However, the code itself is clean and well structured.
>
> We appreciate the reader's advice.
> We will add a README file with explanations of the code's APIs, and host the code on GitHub for the camera-ready version.
>
> ## Punctuation at the end of paragraph header is missing in some cases, and consistently in the Appendix.
>
> We appreciate the correction; we missed the inconsistent punctuation in \paragraph{} tags entirely in our pre-submission reviews.
>
> ## Figures in page 8 are missing numbers, and they should be referenced in the manuscript.
>
> We are grateful for the reviewer's attention to detail, particularly their highlighting of specific issues in the formatting of Section 5.
> As mentioned in the introduction, we will revise Section 5 in our next draft, including adding the tables, simplifying the diagrams, and better describing their contents.
>
> ## Conclusion
>
> We appreciate the reviewer's recognition of the clinical motivations for designing a new scoring rule approach to capture distribution shift under asymmetric costs.
> They also encouraged us to clarify that our approach relies on being able to bound prevalence at evaluation time, and they suggested that we expand and clarify our empirical validation with more real world examples in which bounded DCA log score outperforms the most widely used metrics.
> We thank the reviewer for these suggestions, which we have implemented as described above.
>
>
> [1] Epstein, E. S. (1969). A scoring system for probability forecasts of ranked categories. Journal of Applied Meteorology and Climatology, 8(6), 985-987.
>
> [2] Hyvärinen, A. (2005). Estimation of non-normalized statistical models by score matching. Journal of Machine Learning Research, 6, 695-709.
>
> [3] Matheson, J. E., & Winkler, R. L. (1976). Scoring rules for continuous probability distributions. Management Science, 22(10), 1087-1096.
>
> [4] Johansen, S. (1974). The extremal convex functions. Mathematica Scandinavica, 34(1), 61-68.
>
> [5] Zimmerman, J. E., Kramer, A. A., McNair, D. S., & Malila, F. M. (2006, May). Acute physiology and chronic health evaluation (APACHE) IV: Hospital mortality assessment for today's critically ill patients. Critical Care Medicine, 34(5), 1297-1310.
>
> [6] Pollard, T. J., Johnson, A. E. W., Raffa, J. D., Celi, L. A., Mark, R. G., & Badawi, O. (2018). The eICU collaborative research database, a freely available multi-center database for critical care research. Scientific Data, 5, 180178.
>
> [7] Johnson, A., Pollard, T., Badawi, O., & Raffa, J. (2021). eICU collaborative research database demo (version 2.0.1). PhysioNet.

---

> > ### Comment · Reviewer_syGE · 2025-08-05
> >
> > Table 1 currently presents a mix of discrimination and calibration metrics, where the interpretation of values varies (some metrics are better when higher, others when lower). To improve clarity, the authors could consider adding directional indicators (e.g., ↑ for "higher is better", ↓ for "lower is better") next to each metric. This would help readers quickly understand the desired direction of performance for each metric and avoid confusion.

---

### Official Review · Reviewer_SEhd · 2025-06-28

**Clarity:** 3
**Significance:** 2
**Originality:** 3
**Rating:** 5
**Confidence:** 3

**Summary:**

This paper is a theoretical paper motivated by clinical problems. In particular, it proposed an evaluation metric for binary classifiers in clinical decision support, explicitly addressing three key clinical concerns: calibration, robustness to label shift, and asymmetric error costs. Leveraging the Schervish representation of proper scoring rules, the authors derive a cost-sensitive, clipped variant of cross-entropy—termed the DCA log score—that averages performance over a bounded range of plausible class prevalences. A subgroup analysis using the eICU dataset illustrates how traditional metrics may obscure disparities that the proposed framework reveals.

**Questions:**

1. Comparative Evaluation: Have you conducted empirical comparisons between your proposed DCA log score and existing cost-sensitive or calibration-aware metrics such as expected calibration error (ECE)? It would be helpful to understand how your metric performs in model selection or ranking tasks relative to these established alternatives.

2. Figure Clarity (Section 5): The figure in Section 5 is difficult to interpret. Specifically, Subfigure 1 includes two labels marked “B,” and Subfigure 2 contains two “A”s and two “B”s. The current caption does not adequately explain what each label refers to.

**Ethical Concerns:**

["NO or VERY MINOR ethics concerns only"]

**Final Justification:**

The authors have addressed most of my concerns.

**Limitations:**

yes.

**Quality:**

3

**Strengths And Weaknesses:**

Strengths:
1. High Relevance: The paper addresses an important and underexplored challenge in medical ML—namely, that commonly used metrics (e.g., accuracy and AUC-ROC) often fail to align with key clinical priorities such as calibration under distribution shift and sensitivity to asymmetric error costs.
2. Strong Theoretical Foundation: The authors build on the Schervish representation to derive a bounded variant of the DCA log score. This derivation effectively bridges calibration and cost-sensitive thresholding under uncertainty, yielding a theoretically grounded and interpretable evaluation framework.

Weaknesses:
1. Limited Novelty Relative to Prior Work: While the clinical framing is timely and well-motivated, much of the theoretical basis—including proper scoring rules, label shift correction, and cost-sensitive evaluation—has been explored in prior literature. The main contribution lies in adapting and synthesizing these ideas for the clinical evaluation context.
2. Narrow Empirical Scope: The empirical analysis is limited to a single illustrative case. While the example is helpful, more extensive benchmarking across multiple datasets and models would provide stronger evidence for the practical advantages of the proposed metric over standard alternatives.

---

> ### Author Rebuttal · Authors · 2025-07-29
>
> # Introduction: New Analysis and Major Commitments for Revision
>
> We thank the reviewers for thoughtfully highlighting the value of more explicit comparisons of our proposed metric to popular alternatives on real data.  To that end, we have extended the analysis from Section 5 on the performance of the Apache IV mortality prediction algorithm in different subgroups of the publicly available eICU dataset, and will include the following tables in the revised draft.
>
> | Metric | Caucasian | African-American |
> |--------|-----------|------------------|
> | Bounded DCA Log Score (calibration-only) | **0.999** | 0.927 |
> | ECE | **0.829** | 0.889 |
> | Bounded DCA Log Score (discrimination-only) | 0.957 | **0.973** |
> | AUC-ROC | 0.868 | **0.907** |
> | Bounded DCA Log Score | **0.956** | 0.900 |
> | ECE & AUC-ROC | ? | ? |
>
> *Table 1: Calibration and Discrimination for Caucasian and African-American Patients*
>
> Both commonly used and proposed calibration measures agree that the model has better calibration for African-American patients than Caucasian, but worse discrimination.
> However, the calibration and discrimination components for Bounded DCA Log Score can be linearly combined, while there is no principled way to combine AUC-ROC with ECE to reach an overall judgment.
>
> | Metric | Caucasian | African-American |
> |--------|-----------|------------------|
> | Accuracy Curve@1% Prevalence | **0.990** | 0.978 |
> | Accuracy Curve@5% Prevalence | **0.953** | 0.896 |
> | Accuracy Curve@10% Prevalence | **0.912** | 0.793 |
> | Bounded DCA Log Score | **0.956** | 0.900 |
> | Accuracy | 0.922 | **0.961** |
>
> *Table 2: Accuracy for Caucasian and African-American patients at varying prevalences*
>
> The most powerful analysis technique here is to graph accuracy at each prevalence and compare the two entire curves visually. As the table suggests, Apache IV on the public eICU data performs better for Caucasian patients than for African-American patients at any given prevalence.
> Comparing two Bounded DCA Log Scores averages the heights of these curves and the scalar summary preserves their ranking.
> The empirical accuracy for each subgroup is measured at different prevalences.
>
> | Metric | Male | Female |
> |------------|------|--------|
> | Accuracy@1% | 0.989 | 0.990 |
> | Accuracy@5% | 0.948 | 0.954 |
> | Accuracy@25% | 0.818 | 0.814 |
> | Accuracy@75% | 0.846 | 0.832 |
> | Accuracy@95% | 0.953 | 0.953 |
> | Accuracy@99% | 0.990 | 0.990 |
>
> *Table 3: Accuracy for Male and Female Patients*
>
> Apache IV performs better for female patients than male patients at low mortality prevalences, worse for roughly balanced mortality, and about the same at high mortality prevalences. However, only the low prevalence ranges are clinically relevant.
>
> | Metric | Male | Female |
> |--------|------|--------|
> | Brier | 0.062 | 0.062 |
> | Cross-Entropy / NLL | 0.214 | 0.218 |
> | Bounded DCA Log Score | 0.917 | 0.926 |
>
> *Table 4: Scoring Rules for Male and Female Patients*
>
> Although unbounded Brier score and cross-entropy are related to integrals of net cost, and are therefore related to net benefit, they give misleading answers because they put too much weight at high, clinically unrealistic prevalences.
>
>
>
> # Specific Reviewer Questions and Replies
>
> ## Limited Novelty Relative to Prior Work:
> > While the clinical framing is timely and well-motivated, much of the theoretical basis—including proper scoring rules, label shift correction, and cost-sensitive evaluation—has been explored in prior literature.
> The main contribution lies in adapting and synthesizing these ideas for the clinical evaluation context.
>
> We appreciate the reviewer's recognition of this work's progress towards synthesizing proper scoring rules, evaluating classifiers under unknown label shift, and providing frameworks for cost-sensitive evaluation.
> While label shift correction and cost-sensitive learning are well developed fields, evaluation methods to adapt to them are less widespread; for example, a recent review of 173 papers using cost-sensitive learning in medicine found less than 10\% of papers actually used existing cost-sensitive metrics [1].
>
> We wish to emphasize that, to our knowledge, this is the first framework to combine these three dimensions into a unified, practically deployable metric, directly addressing the persistent challenge in clinical ML of actionable model selection under label shift and asymmetric costs.
>
> | Metric | Domain Adaptation Failure | Cost-Sensitive Evaluation Failure | Decomposability Failure | Notes |
> |--------|---------------------------|-----------------------------------|-------------------------|-------|
> | **Accuracy** | Does not handle subgroup prevalence shifts. | Assumes symmetric costs (FP = FN) | Does not reflect calibration. Very little information on discrimination. | Simpson's paradox effects when subgroup prevalences differ |
> | **AUC-ROC** | Invariant to prevalence | Range of costs chosen by model, not clinicians | Pure discrimination, no calibration | Can't combine with (e.g.) ECE |
> | **Expected Calibration Error (ECE)** | Domain-specific adaptation unclear | No cost sensitivity | Pure calibration, no discrimination | Can't combine with (e.g.) AUC-ROC |
> | **Brier Score** | Unrealistic prevalence ranges (e.g. half of weight above 50% mortality)| Cannot combine asymmetric costs with imbalanced classes | Decomposes into calibration and discrimination | Misleading at clinical prevalences |
> | **Cross-Entropy / NLL** | Unrealistic prevalence ranges (e.g. half of weight above 50% mortality) | Cannot combine asymmetric costs with imbalanced classes | Decomposes into calibration and discrimination | Misleading at clinically relevant prevalences |
> | **Net Benefit** | Requires known prevalence | Requires known costs | No decomposition into calibration and discrimination | Requires exact knowledge |
>
> We will clarify the practical implications of this synthesis in the revision; we will also expand Section 5 to include the additional empirical tables and guided figure walkthroughs outlined in the introduction.
>
> ## Narrow Empirical Scope:
> > The empirical analysis is limited to a single illustrative case.
> While the example is helpful, more extensive benchmarking across multiple datasets and models would provide stronger evidence for the practical advantages of the proposed metric over standard alternatives.
>
> We have added new analyses of the Apache IV mortality prediction (Tables 1-4 above).
> This real clinical dataset on the paradigmatic task of ICU mortality serves to illustrate a typical medical task with low but subgroup-heterogeneous prevalence.
> By carefully studying this dataset, we were able to find subtle but important effects, such as the Simpson's paradox-style effect of subgroup prevalence on measured subgroup accuracy and  Apache IV's stronger performance for women at clinically relevant prevalences.
> Importantly, these effects were masked by worse performance at high prevalence, and we show that our metrics captured those.
> The new analysis finds further examples where existing metrics make the wrong decision.
> We commit to including tables and clear descriptions of these analyses in the next version.
>
> ## Comparative Evaluation:
> > Have you conducted empirical comparisons between your proposed DCA log score and existing cost-sensitive or calibration-aware metrics such as expected calibration error (ECE)? It would be helpful to understand how your metric performs in model selection or ranking tasks relative to these established alternatives.
>
> We thank the reviewer for suggesting these improvements
>
> -  We've added ECE in the new analysis (Table 1, above); this demonstrates a clear clinical example where the comparison in ECE gives one answer, but AUC-ROC gives a different one.
> The critical advantage of DCA log score is that the calibration and discrimination components can be linearly combined, producing a single scalar score to guide decisions, where ECE cannot be combined with (e.g.) AUC-ROC.
> - Table 2 shows a real clinical example where the classifier performs better on one subgroup than another at any relevant prevalence, but measuring accuracy gives the wrong answer.
> This occurs because prevalences in the evaluation data are different from prevalences in the training data.
> - Table 3 and 4 show a real clinical example where one subgroup has better performance at clinically relevant prevalences, but the effect reverses at unrealistically high prevalences.
> This leads the unbounded Brier score and cross-entropy to give the wrong results.
>
> ## Figure Clarity (Section 5):
> > The figure in Section 5 is difficult to interpret.
> Specifically, Subfigure 1 includes two labels marked “B,” and Subfigure 2 contains two “A”s and two “B”s.
> The current caption does not adequately explain what each label refers to.
>
> We appreciate the reviewer pointing out this error -- we will use unique labels in the next revision, rather than repeating letters in different colors.
> Further, we will revise the caption to more clearly walk through the quantities represented by various vertical intervals on the graph.
>
> ## Conclusion
>
> We appreciate the reviewer's recognition of the clinical motivations for designing a new scoring rule approach to capture distribution shift under asymmetric costs.
> Their encouragment has helped us commit to expanding Section 5 with further empirical comparison of the Bounded DCA Log Score to the most widely adopted metrics, focusing on the narrative description of where and how the metric captures different aspects of model performance, and thereby justifying its use.
>
> [1] Araf, I., Idri, A., & Chairi, I. (2024, March). Cost-sensitive learning for imbalanced medical data: A review. Artificial Intelligence Review, 57(4), 80.

---

> > ### Comment · Reviewer_SEhd · 2025-08-01
> >
> > I appreciate the effort the authors have made for clarification and providing additional simulations results. I have increase my score to 5.

---

### Official Review · Reviewer_RD77 · 2025-06-30

**Clarity:** 3
**Significance:** 4
**Originality:** 2
**Rating:** 4
**Confidence:** 4

**Summary:**

This work focuses on the mismatch between model evaluation and clinical priorities, specially calibration and robustness to distributional shifts. This work characterizes where existing methods, like accuracy and AUROC, fail, and propose an evaluation framework for selecting calibrated thresholded classifiers that can handle bounded distributional shift and domain-specific cost asymmetries. To do so, the authors propose a variant of the cross-entropy score using clipping that averages cost-weighted performance over real ranges of distributional label shift.

**Questions:**

1. How do brier score and expected calibration error compare to the proposed evaluation metric towards the goal of measuring calibration and providing an interpretable metric real-world deployment?
2. How can we apply the new evaluation metric to real clinical data, and how can the results be interpreted in a meaningful way?
3. How would evaluation in real clinical data differ between existing measures of calibration and the proposed clipped log metric?
4. Can the brier score be used in a similar context to the log cross entropy, as the brier score is also a proper scoring rule that can be decomposed into sharpness and calibration?

**Ethical Concerns:**

["NO or VERY MINOR ethics concerns only"]

**Final Justification:**

Author response helped clarify major reasons for rejection, impact of proposed metric is still of concern.

**Limitations:**

yes

**Quality:**

3

**Strengths And Weaknesses:**

**Strengths**

1. The work approaches an important problem and correctly identifies that many metrics that we use today (i.e., accuracy and AUROC) are incompatible with clinical needs.
2. A solution which allows clinical input is useful, as domain experts provide a better opportunity to know what type of label/distributional shift is expected.
3. The clipped cross-entropy loss is a nice extension of a commonly used loss, and the provides a nice interpretation with respect to net-benefit over multiple class prevalences.
4. The authors do a good job surveying past work in limitations of brier score and expected calibration error, which are often used as measures of calibration in the clinical space but can often not be extended to asymmetric costs or ranges of different prevalences.

**Weaknesses**

1. The findings of accuracy and AUROC with respect to misalignment with some clinical objectives is not particularly novel and is well known in the clinical space. This is especially true in the very related space of survival analysis, where recent work has moved beyond simply reporting the C-index (an analog of AUROC) and focused more on calibration. Hence, many of the results showing the limitations of AUROC/accuracy are not particularly interesting findings when placed in the broader context of the clinical space. Instead, the paper should focus more time showing why existing measures of calibration in the clinical space (i.e., NLL, brier score, expected calibration error (ECE)) are not sufficient.
2. Similar to the point above, the experimental results do not provide significant findings for this paper. As noted, accuracy and AUROC are already well-known in their limitations for calibration and insensitivity to class balance and label shift. It would be more useful to provide an apples to apples comparison of existing calibration metrics in the clinical space (brier score, ECE) with the proposed loss to show how theoretical guarantees manifest and how interpretations differ. The current analysis does not help in showing how the proposed evaluation metric can be applied nor how it should be interpreted in real clinical scenarios.

---

> ### Author Rebuttal · Authors · 2025-07-29
>
> # Introduction: New Analysis and Major Commitments for Revision
>
> We thank the reviewers for thoughtfully highlighting the value of more explicit comparisons of our proposed metric to popular alternatives on real data.  To that end, we have extended the analysis from Section 5 on the performance of the Apache IV mortality prediction algorithm in different subgroups of the publicly available eICU dataset, and will include the following tables in the revised draft.
>
> | Metric | Caucasian | African-American |
> |--------|-----------|------------------|
> | Bounded DCA Log Score (calibration-only) | **0.999** | 0.927 |
> | ECE | **0.829** | 0.889 |
> | Bounded DCA Log Score (discrimination-only) | 0.957 | **0.973** |
> | AUC-ROC | 0.868 | **0.907** |
> | Bounded DCA Log Score | **0.956** | 0.900 |
> | ECE & AUC-ROC | ? | ? |
>
> *Table 1: Calibration and Discrimination for Caucasian and African-American Patients*
>
> Both commonly used and proposed calibration measures agree that the model has better calibration for African-American patients than Caucasian, but worse discrimination.
> However, the calibration and discrimination components for Bounded DCA Log Score can be linearly combined, while there is no principled way to combine AUC-ROC with ECE to reach an overall judgment.
>
> | Metric | Caucasian | African-American |
> |--------|-----------|------------------|
> | Accuracy Curve@1% Prevalence | **0.990** | 0.978 |
> | Accuracy Curve@5% Prevalence | **0.953** | 0.896 |
> | Accuracy Curve@10% Prevalence | **0.912** | 0.793 |
> | Bounded DCA Log Score | **0.956** | 0.900 |
> | Accuracy | 0.922 | **0.961** |
>
> *Table 2: Accuracy for Caucasian and African-American patients at varying prevalences*
>
> The most powerful analysis technique here is to graph accuracy at each prevalence and compare the two entire curves visually. As the table suggests, Apache IV on the public eICU data performs better for Caucasian patients than for African-American patients at any given prevalence.
> Comparing two Bounded DCA Log Scores averages the heights of these curves and the scalar summary preserves their ranking.
> The empirical accuracy for each subgroup is measured at different prevalences.
>
> | Metric | Male | Female |
> |------------|------|--------|
> | Accuracy@1% | 0.989 | 0.990 |
> | Accuracy@5% | 0.948 | 0.954 |
> | Accuracy@25% | 0.818 | 0.814 |
> | Accuracy@75% | 0.846 | 0.832 |
> | Accuracy@95% | 0.953 | 0.953 |
> | Accuracy@99% | 0.990 | 0.990 |
>
> *Table 3: Accuracy for Male and Female Patients*
>
> Apache IV performs better for female patients than male patients at low mortality prevalences, worse for roughly balanced mortality, and about the same at high mortality prevalences. However, only the low prevalence ranges are clinically relevant.
>
> | Metric | Male | Female |
> |--------|------|--------|
> | Brier | 0.062 | 0.062 |
> | Cross-Entropy / NLL | 0.214 | 0.218 |
> | Bounded DCA Log Score | 0.917 | 0.926 |
>
> *Table 4: Scoring Rules for Male and Female Patients*
>
> Although unbounded Brier score and cross-entropy are related to integrals of net cost, and are therefore related to net benefit, they give misleading answers because they put too much weight at high, clinically unrealistic prevalences.
>
>
>
> # Specific Reviewer Questions and Replies
>
> We thank the reviewer for their helpful comments.
> We appreciate that the reviewer agrees that this is an important problem, that clipped cross-entropy is a nice extension, and that allowing clinician input is useful.
>
> ## Question 1: How do brier score and expected calibration error compare to the proposed evaluation metric towards the goal of measuring calibration and providing an interpretable metric real-world deployment?
>
> ### Expected Calibration Error
> ECE has flaws as an estimator of calibration, and there is no principled way to combine it with measures of discrimination (e.g. AUC-ROC minus ECE doesn't mean anything).
>
> 1. **Binning causes statistical difficulties**
>
> There is an intrinsic bias-variance tradeoff with binning [1]:
> - if the bins are small, the variance of the average label is high, and this biases the estimate up.
> Even with thousands of samples and perfect calibration, [2] shows this can produce large overestimates of miscalibration.
> - if the bins are large, miscalibrations can cancel out, biasing the estimate down.
> [3] dramatizes this by showing that in the limiting case of 1 bin, guessing 0% or 100% randomly at the population base rate gives ECE of 0.
>
> 2. **ECE doesn't combine with measures of discrimination**
>
> It is commonly recommended to report both ECE and AUC-ROC which, as mentioned in Section 3, provides a pure measure of discrimination with no calibration component.
> Unfortunately, there is no principled way to combine the two numbers to make a joint decision.
> For example, subtracting ECE from AUC-ROC has no interpretation, nor can ECE be combined with other discrimination measures.
> Table 1 shows an example where this means we cannot decide which subgroup has better performance.
> The calibration and discrimination components of Bounded DCA log score can, in contrast, be easily combined.
>
> ## Brier Score
>
> Comparisons that use a metric ought to indicate when the model does better on one population than another on average across the clinically relevant range of prevalences.
> This requires us to restrict the range of prevalences, which Brier score does not do.
> Our contribution is to:
> a) show when and why scoring rules can be interpreted as distributions of net benefit over varying prevalences.
> b) show how to restrict the range of prevalences using a simple analytic formula.
> c) show why the clipping the log score is more useful than clipping the Brier score.
>
> 1. **Why Clipping?**
>
> The two concrete differences between Brier score and our metric are that our metric introduces bounds on the prevalence and that it also allows for asymmetric costs.
> Clincians typically do not know actual deployment prevalence at the time they choose a model, but they are often able to place useful constraints on prevalence, such as "less than half." See Table 3 and Table 4 for a real clinical illustration where this is important.
>
> 2. **Why DCA Log Score and not DCA Brier score?**
>
> This overlaps with Question 4, so we will address it in greater detail there.
>
> # Question 4: Can the brier score be used in a similar context to the log cross entropy, as the brier score is also a proper scoring rule that can be decomposed into sharpness and calibration?
>
> Yes, the Brier score can also be clipped, but we show that cross-entropy makes more sense in this context.
>
> 1. Cross-entropy reflects what we mean by an uncertainty interval that covers orders of magnitude, as we discuss in section 4.2 (lines 290 to 295).
>
> To give a concrete example, the range of syphilis prevalence in US states is between 2 in 1,000 and 2 in 100,000, depending on the state.
> We believe that the typical practitioner would expect the median of this interval to be 2 in 10,000.
> Brier score, by contrast, favors prevalences close to 1/2, so the median of the interval will be 1 in 1,000.
>
> 2. The formulation of cross-entropy is readily adapted to a restricted range of prevalences under asymmetric costs.
>  As we show in Theorem D.7, the Brier score can be adapted to restrict the range of prevalences if symmetric costs are assumed.
> Adding asymmetric costs produces an unwieldy expression, reducing intepretability.  We will include it in the appendix for completeness.
>
> # Question 2: How can we apply the new evaluation metric to real clinical data, and how can the results be interpreted in a meaningful way?
>
> We appreciate the reviewer prompting us to clarify how we interpret ranking differences among metrics for real clinical data.
> We have applied the evaluation metric to real mortality predictions from real ICU data [4] in Section 5 and added new analyses in Tables 1 to 4 of this rebuttal.
> When it is possible to a detailed analysis, we think visualizing accuracy as a function of prevalence provides intuition for why metrics differ.
> However, we believe the mark of a good scalar metric is that we can simply compare magnitudes; in the example in Section 5, the DCA log score is simply the average accuracy across the clinically relevant range of prevalences, and the examples show that comparing magnitudes is valid.
>
> # Question 3: How would evaluation in real clinical data differ between existing measures of calibration and the proposed clipped log metric?
>
> We thank the reviewer for prompting us to explain why the bounded DCA log score gives different results on the ICU data from existing metrics.
> The introduction provides concrete examples:
>
> - When one subgroup has higher calibration and another has higher discrimination, it is possible to answer the question "which is doing better?" by a linear combination of the scores.
> - When subgroups have different prevalence, it is still possible to identify *where* the classifier is performing better.
> - Evaluation can be restricted to clinically meaningful ranges of prevalence.
>
> ## Conclusion
>
> We appreciate the reviewer's recognition of the clinical motivations for designing a new scoring rule approach to capture distribution shift under asymmetric costs.
> Similarly, we are grateful for the encouragement to expand our empirical validation to show real world examples in which bounded DCA log score outperforms the most widely used metrics (i.e. ECE, AUC-ROC, Accuracy, Brier score and cross-entropy / NLL).
>
>
> [1] Nixon, J., Dusenberry, M. W., Zhang, L., Jerfel, G., & Tran, D. (2019, June). Measuring calibration in deep learning. CVPR Workshops.
>
> [2] Roelofs, R., Cain, N., Shlens, J., & Mozer, M. C. (2022, March). Mitigating bias in calibration error estimation. AISTATS (Vol. 151, pp. 4036-4054).
>
> [3] Ferrer, L., & Ramos, D. (2024). Evaluating posterior probabilities: Decision theory, proper scoring rules, and calibration. arXiv:2408.02841.
>
> [4] Johnson, A., Pollard, T., Badawi, O., & Raffa, J. (2021). eICU collaborative research database demo (version 2.0.1). PhysioNet.

---

> > ### Comment · Reviewer_RD77 · 2025-08-05
> >
> > I appreciate the author's response to my comment as they help clarify many of my questions. I really appreciate the new results as they directly address some of my major limitations.
> >
> > My main remaining concern regards the utility of the proposed metric over using separate existing measures of calibration and discrimination. A justification for the benefit of the DCA log score is that it can linearly combine calibration and discrimination for accurate judgements. However, in most clinical scenarios, an end-user might way to trade-off between calibration and discrimination performance based on priorities of the task. For example, the end-user might weight calibration more than discrimination in cases where point estimates are more important, vs. weighting discrimination more when ranking is all that matters. This situation is the norm in clinical tasks. In these situations, having the scores interpretable yet separate might make more sense. Can the proposed metric still have utility in these settings?
> >
> > Despite my concern above, I have raised my score to a 4 due to the author response.

---

> > > ### Author Response · Authors · 2025-08-06
> > >
> > > We thank the reviewer for the insightful question about practitioners' needs for task-dependent trade-offs between calibration and discrimination.
> > > While we addressed this in Section 5 and Appendix H, we appreciate the encouragement to foreground this trade-off more fully in our exposition.
> > >
> > > The core difficulty in comparing AUC-ROC and ECE is that they live on different scales, so they are only qualitatively comparable.
> > > If a user wishes to place, say, 10% more weight on discrimination than calibration, there is no principled default for equal weighting to begin with.
> > > Proper scoring rules provide such a combination strategy, and our proposed metric in particular expresses both components in units of true positives (Section 4.3).
> > > From that baseline, a user can then increase the weight on either component by, e.g., 10%.
> > >
> > > We consider in more depth the logical extremes of the trade-off: point estimates and ranking.
> > >
> > > # Point Estimates
> > >
> > > Consider the ACC/AHA guidelines for primary prevention of cardiovascular disease.
> > > They recommend prescribing statins based on 10-year cardiovascular disease risk, thresholded at 2.5%, 5%, 7.5%, and 20% [1].
> > > A calibration-only approach can be pathological: a model that predicts the population base risk (around 10%) for everyone would receive an excellent score from calibration-only metrics like ECE [2], yet it would be clinically unhelpful because it recommends giving everyone a light dose of statins.
> > > For such point-estimate settings, combining calibration with discrimination is preferable, which is exactly what our metric does.
> > >
> > > # Ranking
> > > Kidney allocation decisions are intrinsically ranking-based [3].
> > > Because there are more candidates than available kidneys, the absolute calibration of end-stage renal disease (ESRD) mortality predictions is secondary to ordering candidates.
> > > In these settings, ranking metrics differ in how they weight performance at different parts of the list [4].
> > >
> > > As we show in Theorem 3.2, even if a model is not selected primarily for calibration, AUC-ROC has a clear interpretation when the model is well-calibrated: it corresponds to picking thresholds uniformly from the two classes in the training data.
> > > By contrast, our proposed metric allows users to emphasize the regions of the ordering that matter most for the task.
> > >
> > > For example, if the median transplant recipient has an Estimated Post-Transplant Survival (EPTS) around 36% [3], our metric can focus evaluation on the neighborhood of that value.
> > >
> > > # Conclusion
> > > The key contribution of proper scoring rules to navigating calibration–discrimination trade-offs is to provide a principled default combination strategy.
> > > This is typically the right place to start—even in seemingly simple point-estimate cases—after which a user can adjust weights if desired.
> > > We thank the reviewer for highlighting this trade-off; we will elevate this discussion from Appendix H to a more prominent position in the main text.
> > >
> > > # References
> > >
> > > [1] D. K. Arnett et al., "2019 ACC/AHA Guideline on the Primary Prevention of Cardiovascular Disease," Circulation, 140(11): e596–e646, 2019.
> > >
> > > [2] K. L. King et al., "Characterization of Transplant Center Decisions to Allocate Kidneys to Candidates With Lower Waiting List Priority," JAMA Network Open, 6(6): e2316936, 2023.
> > >
> > > [3] Ferrer, L., & Ramos, D. (2024). Evaluating posterior probabilities: Decision theory, proper scoring rules, and calibration. arXiv:2408.02841.
> > >
> > > [4] M. B. McDermott et al., "A closer look at AUROC and AUPRC under class imbalance," in NeurIPS '24.

---

### Official Review · Reviewer_75b4 · 2025-07-03

**Clarity:** 4
**Significance:** 3
**Originality:** 3
**Rating:** 5
**Confidence:** 3

**Summary:**

This study presents a new method for measuring the performance of ML predictive models for clinical applications. The method emphasizes measuring performance at deployment and aims to keep the method simple and straightforward (for calculation and interpretation).

**Questions:**

Is Theorem E.1 a proof for 4.1? What's the exact novelty in the formulation, is it only the clipping process or something beyond?

**Ethical Concerns:**

["NO or VERY MINOR ethics concerns only"]

**Final Justification:**

The author's response addresses (and acknowledges) some of my comments.

**Limitations:**

yes

**Paper Formatting Concerns:**

good

**Quality:**

3

**Strengths And Weaknesses:**

**Strengths**
- The study targets an important and critical area in ML for healthcare.
- Extensive background and context are provided prior to presenting the proposed method (mostly Theorem 4.1).
- The study also aims to maintain a pragmatic approach by making the proposed way easier to interpret.

**Weaknesses**
- The study starts with a rather unorthodox introduction, and right off the bat presents “three priorities for clinical purposes.” These are then treated as the foundation of all that follows. While these three are sound and reasonable, they are not an exhaustive list encompassing all issues in this area, as the text seems to imply.
- While the study targets a very critical and timely issue, the main pitch is hardly news to the practitioners in the field. As the study acknowledges, many have pushed for going beyond accuracy and AUROC, yet those measures remain mainstream. Even using simple calibration or net benefit calculations are less frequently seen. Now, one (including this reviewer) may argue that part of the reason for the current situation is that most studies similar to the present work remain very high-level and skip the practical implications and use cases in favor of the theoretical foundations. The current work may appeal to a limited Nuerips audience, but expanding the real-world scenarios (as presented in section 5) is critical to demonstrate the true value of the method.

---

> ### Author Rebuttal · Authors · 2025-07-29
>
> # Introduction: New Analysis and Major Commitments for Revision
>
> We thank the reviewers for thoughtfully highlighting the value of more explicit comparisons of our proposed metric to popular alternatives on real data.  To that end, we have extended the analysis from Section 5 on the performance of the Apache IV mortality prediction algorithm in different subgroups of the publicly available eICU dataset, and will include the following tables in the revised draft.
>
> | Metric | Caucasian | African-American |
> |--------|-----------|------------------|
> | Bounded DCA Log Score (calibration-only) | **0.999** | 0.927 |
> | ECE | **0.829** | 0.889 |
> | Bounded DCA Log Score (discrimination-only) | 0.957 | **0.973** |
> | AUC-ROC | 0.868 | **0.907** |
> | Bounded DCA Log Score | **0.956** | 0.900 |
> | ECE & AUC-ROC | ? | ? |
>
> *Table 1: Calibration and Discrimination for Caucasian and African-American Patients*
>
> Both commonly used and proposed calibration measures agree that the model has better calibration for African-American patients than Caucasian, but worse discrimination.
> However, the calibration and discrimination components for Bounded DCA Log Score can be linearly combined, while there is no principled way to combine AUC-ROC with ECE to reach an overall judgment.
>
> | Metric | Caucasian | African-American |
> |--------|-----------|------------------|
> | Accuracy Curve@1% Prevalence | **0.990** | 0.978 |
> | Accuracy Curve@5% Prevalence | **0.953** | 0.896 |
> | Accuracy Curve@10% Prevalence | **0.912** | 0.793 |
> | Bounded DCA Log Score | **0.956** | 0.900 |
> | Accuracy | 0.922 | **0.961** |
>
> *Table 2: Accuracy for Caucasian and African-American patients at varying prevalences*
>
> The most powerful analysis technique here is to graph accuracy at each prevalence and compare the two entire curves visually. As the table suggests, Apache IV on the public eICU data performs better for Caucasian patients than for African-American patients at any given prevalence.
> Comparing two Bounded DCA Log Scores averages the heights of these curves and the scalar summary preserves their ranking.
> The empirical accuracy for each subgroup is measured at different prevalences.
>
> | Metric | Male | Female |
> |------------|------|--------|
> | Accuracy@1% | 0.989 | 0.990 |
> | Accuracy@5% | 0.948 | 0.954 |
> | Accuracy@25% | 0.818 | 0.814 |
> | Accuracy@75% | 0.846 | 0.832 |
> | Accuracy@95% | 0.953 | 0.953 |
> | Accuracy@99% | 0.990 | 0.990 |
>
> *Table 3: Accuracy for Male and Female Patients*
>
> Apache IV performs better for female patients than male patients at low mortality prevalences, worse for roughly balanced mortality, and about the same at high mortality prevalences. However, only the low prevalence ranges are clinically relevant.
>
> | Metric | Male | Female |
> |--------|------|--------|
> | Brier | 0.062 | 0.062 |
> | Cross-Entropy / NLL | 0.214 | 0.218 |
> | Bounded DCA Log Score | 0.917 | 0.926 |
>
> *Table 4: Scoring Rules for Male and Female Patients*
>
> Although unbounded Brier score and cross-entropy are related to integrals of net cost, and are therefore related to net benefit, they give misleading answers because they put too much weight at high, clinically unrealistic prevalences.
>
>
>
> # Specific Reviewer Questions and Replies
>
> We thank the reviewer for their thoughtful, thorough review.  Their comments brought out many of the issues that got squeezed out by page limits, and we appreciate the feedback and the opportunity to bring back important context.
>
> ## While the study targets a very critical and timely issue, the main pitch is hardly news to the practitioners in the field.
>
> > As the study acknowledges, many have pushed for going beyond accuracy and AUROC, yet those measures remain mainstream. Even using simple calibration or net benefit calculations are less frequently seen. Now, one (including this reviewer) may argue that part of the reason for the current situation is that most studies similar to the present work remain very high-level and skip the practical implications and use cases in favor of the theoretical foundations. The current work may appeal to a limited Nuerips audience, but expanding the real-world scenarios (as presented in section 5) is critical to demonstrate the true value of the method.
>
> It is correct that adoption is the key challenge.
> The math is necessary here because it shows that measuring net benefit and calibration together with one good metric can be easier and more powerful than measuring them separately with two metrics.
> However, the main way we intend to move the needle is by providing practical advantages:
> 1) having a single summary of calibration and discrimination such that the user can simply pick the largest value (less complex than AUC-ROC + ECE, as shown in Table 1)
> 2) restricting to the domain of clinical relevance (more relevant than Brier score or cross-entropy, as shown in Table 3 and 4)
> 3) separating accuracy differences from prevalence differences (see comparison to Accuracy in Table 2)
> 4) allowing uncertainty about exact prevalence (does not require precise knowledge like Net Benefit)
> 5) computational simplicity: while our approach is not computationally trivial, it is nonetheless based on clipping the well-known cross-entropy measure.
>
> ## Question 1: Is Theorem E.1 a proof for 4.1? What's the exact novelty in the formulation, is it only the clipping process or something beyond?
>
> We appreciate the reviewer pointing out this error; the proof of 4.1 is in fact Theorem E.3.
>
> The novelty in the formula itself is applying the clipping process and making it compatible with asymmetric costs.  Combining the two is, in fact, tricky, and it is a major reason that we prefer the log score-based measures over the Brier-based measures.
>
> Our derivation is novel because we are explicit about starting from empirical aggregates of observed data. We do not assume the distributions of positive and negative classes are given.  This clarifies the causal assumptions needed for this to be valid; distribution shift will not always mirror data resampling.
>
> ## Conclusion
>
> We appreciate the reviewer's recognition of the clinical motivations for designing a new scoring rule approach to capture distribution shift under asymmetric costs and their focus on clinical adoption.  Their encouragment has helped us center the importance of crisp examples with clear descriptions using real data, and commit to expanding Section 5 with further empirical comparison of the Bounded DCA Log Score to the most widely adopted metrics, and more emphasis on describing how to use the DCA Log score to resolve issues in subgroup decomposition that are otherwise difficult to tackle.

---

> > ### Comment · Reviewer_75b4 · 2025-08-05
> >
> > I appreciate the authors' response. This is solid work, and the response addresses (and acknowledges) some of the raised concerns. I updated my score to 5.

---

### Decision · Program_Chairs · 2025-09-17

**Decision:**

Accept (poster)

**Comment:**

**(a) Paper summary**
The paper introduces a bounded DCA log score (clipped cross-entropy) that combines calibration and discrimination into one interpretable metric, supports asymmetric costs, and averages results over clinician-specified prevalence ranges, building on proper scoring rules.

**(b) Strengths**
The method is principled, practical, and easy to compute, while addressing calibration under shift and asymmetric costs. The authors added subgroup analyses on eICU data with comparisons to AUC-ROC, ECE, Brier, and NLL, and showed useful accuracy-vs-prevalence diagnostics.

**(c) Weaknesses**
The novelty is mostly in combining and adapting existing ideas, and the experiments are still somewhat limited. The work is restricted to binary outcomes, and there is an ongoing debate about using one combined metric versus reporting separate scores, plus some minor clarity issues.

**(d) Reasons for decision**
Even with modest novelty and limited scope, the paper provides a clear, theory-based evaluation method that matches clinical needs and gives an actionable scalar for model selection. The added experiments, clarified proofs, and commitments for code and clearer presentation address the main concerns, so I recommend acceptance.

**(e) Discussion period**
Reviewers questioned novelty, empirical breadth, clarity of data and figures, prevalence assumptions, and the use of a single metric. The authors clarified proofs, expanded analyses with more tables and comparisons, explained prevalence bounds, and promised code release and figure fixes. Most reviewers raised their scores, so I consider the issues resolved enough for acceptance.